# Neuronal plasticity during motor rehabilitation training after spinal cord injury
Tim Max Emmenegger [1,2], Gergely David[1], Siawoosh Mohammadi [3,4,5,6], Gabriel Ziegler [7,8], Martina F. Callaghan [9], Alan Thompson [10], Karl John Friston[9], Nikolaus Weiskopf [4,9,11], Tim Killeen[1] & Patrick Freund [1,3,9] ✉

Understanding how the injured nervous system adapts to training is key to advancing rehabilitation across neurological disorders. Using spinal cord injury (SCI) as a model of severe motor-sensory disruption, we investigate whether training-induced structural plasticity in the brain is preserved despite ongoing neurodegeneration. Thirty-two healthy controls and 17 chronic SCI patients (SCI > 6 months) undergo training in a bimanual-bipedal computer-controlled motion game for one hour, four times a week, over one month, with longitudinal microstructural MRI at 3 T, including multiparameter mapping and diffusion MRI. All SCI patients exhibit performance improvements over the training period. These improvements are accompanied by spatially and temporally distributed changes in both gray and white matter, encompassing alterations in volumetric and myelin-sensitive MRI markers. SCI patients demonstrate trajectories of training-induced neuroplasticity that are comparable to, and in some cases greater than, those of healthy controls. Our findings highlight a fundamental capacity of the injured nervous system to adapt through rehabilitation, with evidence from SCI patients showing that—despite severe motor and sensory deficits—the brain can undergo learning-related structural changes. These results suggest that the mechanisms of training-induced plasticity observed in SCI may generalize to rehabilitation across a broad spectrum of neurological disorders.

Traumatic spinal cord injury (SCI) is characterized by motor and sensory impairment at and below the level of the lesion. It is now recognized that SCI is followed by remote, progressive central nervous system (CNS) degeneration lasting at least 5 years, with less extensive neurodegeneration linked to better rehabilitation outcomes[1]. Identifying mechanisms that reduce neurodegeneration or enhance neuroplasticity is key to improving SCI outcomes and may reveal broadly applicable principles for rehabilitation across neurological disorders.

In healthy individuals, motor skill learning engages physical and cognitive processes that drive functional and structural plasticity across interconnected brain regions, including the motor cortex, cerebellum, and hippocampus[2–4]. Neuroimaging studies have shown gray (GM) and white matter (WM) adaptations in brain regions engaged by motor training[5]. Such rehabilitation-induced neuroplasticity has been documented across various neurological conditions[6,7]. More specifically, evidence indicates that SCI triggers significant neuroplasticity across multiple levels of the nervous system, offering potential for functional recovery. In particular, studies have demonstrated that spinal neural circuitry exhibits remarkable automaticity and plasticity following injury, with the capacity to perform complex motor tasks despite disrupted supraspinal input[8]. Plasticity occurs at cerebral cortical, brainstem, spinal cord, and peripheral levels, with mechanisms varying by timeframe: acute changes involve unmasking latent synapses

[1]Spinal Cord Injury Center, Balgrist University Hospital, University of Zurich, Zurich, Switzerland. [2]High Field MR Center, Department of Biomedical Imaging and Image-Guided Therapy, Medical University of Vienna, Vienna, Austria. [3]Max Planck Research Group MR Physics, Max Planck Institute for Human Development, Berlin, Germany. [4]Department of Neurophysics, Max Planck Institute for Human Cognitive and Brain Sciences, Leipzig, Germany. [5]Department of Neuroradiology, University of Lübeck, Lübeck, Germany. [6]Department of System Neuroscience, University Medical Centre Hamburg-Eppendorf, Hamburg, Germany. [7]Institute of Cognitive Neurology and Dementia Research, Otto-von-Guericke-University Magdeburg, Magdeburg, Germany. [8]German Center for Neurodegenerative Diseases (DZNE), Magdeburg, Germany. [9]Department of Imaging Neuroscience, UCL Queen Square Institute of Neurology, University College London, London, UK. [10]Department of Neuroinflammation, UCL Institute of Neurology, University College London, London, UK. [11]Felix Bloch Institute for Solid State Physics, Faculty of Physics and Earth System Sciences, Leipzig University, Leipzig, Germany. ✉e-mail: patrick.freund@balgrist.ch

through neurotransmitter modulation, while chronic adaptations include synaptic efficacy changes and axonal sprouting[9]. Recent systematic review evidence confirms that SCI patients experience plastic changes both spontaneously and through specific neurorehabilitation training, with exercise rehabilitation promoting WM plasticity and increased myelin content[10]. These plasticity changes have been observed in motor and non-motor learning brain regions[7,11,12] and were quantified using metrics, such as myelin water fraction[11], fractional anisotropy (FA)[12], and volumetric analyses[7]. However, although training-induced volumetric neuroplastic changes have been reported[6,7], microstructural changes in myelin or axonal integrity, as quantified by advanced quantitative magnetic resonance imaging (qMRI) techniques, remain rarely explored.

Advanced qMRI techniques, such as diffusion tensor imaging (DTI)[13] and multiparameter mapping (MPM)[14] facilitates noninvasive estimation of myelin and iron content proxies, alongside volumetric changes in GM and WM. These approaches have shown that motor skill learning induces region- and task-specific adaptations, with early changes often emerging in motor-related pathways[5,15]. Neuroplastic responses vary depending on the type of task and the limb involved, reflecting the somatotopic organization of motor networks[16,17]. These training-associated changes involve reorganization of WM tracts and alterations in tissue microstructure, potentially linked to myelin and axonal properties[18,19]. Collectively, these findings support the view that structural plasticity follows a dynamic and hierarchical trajectory during motor learning.

This study employs longitudinal MRI assessments in SCI patients before, during, and after rehabilitation training to: (i) characterise the spatial and temporal evolution of training-induced neuroplastic changes in cortical and subcortical regions involved in motor skill learning, (ii) investigate the relationship between structural neuroplasticity and functional

improvements, and (iii) examine somatotopic adaptations specific to upper and lower limb training. This approach aims to clarify the mechanisms of rehabilitation-induced neuroplasticity in SCI, informing targeted interventions to improve recovery outcomes.

## Results

### Behavioral improvements during a bimanual-bipedal motion game

During the training over 28 days, SCI trainees (Table 1) showed significant improvements in both percent correct stimulus responses (%CSR) ($p = 0.003$, $z = 3.621$) and response time (RT) ($p = 0.005$, $z = 2.817$) (Fig. 1E). Median %CSR increased by 35.9 percentage points from a baseline of 47.8%, reaching a plateau after ~30 days (time = $3/\gamma$ for 95% improvement, Fig. 1E and Supplementary Table 1). RT improved by a median of 26.31 ms from a baseline of 67.11 ms, plateauing after ~38 days (Fig. 1E and Supplementary Table 1). At baseline, SCI upper-limb trainees had higher %CSR than lower-limb trainees, but no RT differences were found. However, the degree of improvement ($\delta$) and rate of change ($\gamma$) did not differ between SCI subgroups for either outcome (Fig. 1F and Supplementary Table 1). Relative to healthy trainees, SCI participants had lower baseline %CSR and slower RT but showed greater %CSR $\delta$ and longer $\gamma$ values for both %CSR and RT (Fig. 1E and Supplementary Table 1). Training gains were symmetric across sides (%CSR:$p > 0.387$; RT: $p > 0.102$) and remained stable from day 28 to 84 (%CSR:$p > 0.058$; RT:$p > 0.086$; Supplementary Table 1).

### Baseline neurodegeneration and training-induced structural neuroplasticity

qMRI revealed widespread baseline structural deficits in SCI patients vs. controls, including reduced GM and WM volume, magnetization transfer

## Table 1 | Patient demographics

| Training group | Subject ID | Age (years) | Time since injury (months) | Tetraplegic/paraplegic | AIS score | Lesion level | Completeness |
|---|---|---|---|---|---|---|---|
| SCI-UL trainees | SCI-UL-01 | 26–30 | 9 | Tetra | A | C4 | Complete |
| SCI-UL trainees | SCI-UL-02 | 56–60 | 73 | Tetra | D | C4 | Incomplete |
| SCI-UL trainees | SCI-UL-03 | 21–25 | 13 | Tetra | A | C7 | Complete |
| SCI-UL trainees | SCI-UL-04 | 66–70 | 7 | Tetra | C | C4 | Incomplete |
| SCI-UL trainees | SCI-UL-05 | 26–30 | 66 | Tetra | D | C6 | Incomplete |
| SCI-UL trainees | SCI-UL-06 | 41–45 | 248 | Tetra | C | C4 | Incomplete |
| SCI-UL trainees | SCI-UL-07 | 31–35 | 31 | Tetra | A | C5 | Complete |
| SCI-UL trainees | SCI-UL-08 | 66–70 | 168 | Tetra | D | C6 | Incomplete |
| SCI-UL trainees | SCI-UL-09 | 61–65 | 287 | Tetra | A | C7 | Complete |
| SCI-LL trainees | SCI-LL-01 | 51–55 | 69 | Para | A | T12 | Complete |
| SCI-LL trainees | SCI-LL-02 | 56–60 | 47 | Para | D | T12 | Incomplete |
| SCI-LL trainees | SCI-LL-03 | 41–45 | 169 | Tetra | D | C3/4; C7/T1 | Incomplete |
| SCI-LL trainees | SCI-LL-04 | 41–45 | 170 | Para | C | L5 | Incomplete |
| SCI-LL trainees | SCI-LL-05 | 45–50 | 25 | Para | D | T12 | Incomplete |
| SCI-LL trainees | SCI-LL-06 | 36–40 | 52 | Para | D | L5 | Incomplete |
| SCI-LL trainees | SCI-LL-07 | 61–65 | 49 | Tetra | D | C3 | Incomplete |
| SCI-LL trainees | SCI-LL-08 | 31–35 | 124 | Para | C | T12 | Incomplete |
| Summary metrics | | | | | | | |
| SCI-UL trainees | Mean ± SD | 46.46 ± 18.94 | 100.22 ± 107.61 | – | – | – | – |
| SCI-LL trainees | Mean ± SD | 47.60 ± 10.04 | 88.13 ± 57.83 | – | – | – | – |
| CON-UL trainees | Mean ± SD | 34.48 ± 10.94 | – | – | – | – | – |
| CON-LL trainees | Mean ± SD | 39.26 ± 13.98 | – | – | – | – | – |
| CON non-trainees | Mean ± SD | 38.89 ± 11.18 | – | – | – | – | – |

Characterization of the patients in terms of training group, 5-year age ranges, AIS score, lesion level, completeness of injury, and time since injury, as well as the mean and standard deviation (SD) of the healthy controls. SCI-UL trainees = patients upper-limb trainees, SCI-LL trainees = patients lower-limb trainees, CON-UL trainees = healthy controls upper-limb trainees, CON-LL trainees = healthy controls lower-limb trainees, CON non-trainees = healthy controls no-trainee.

**Article**

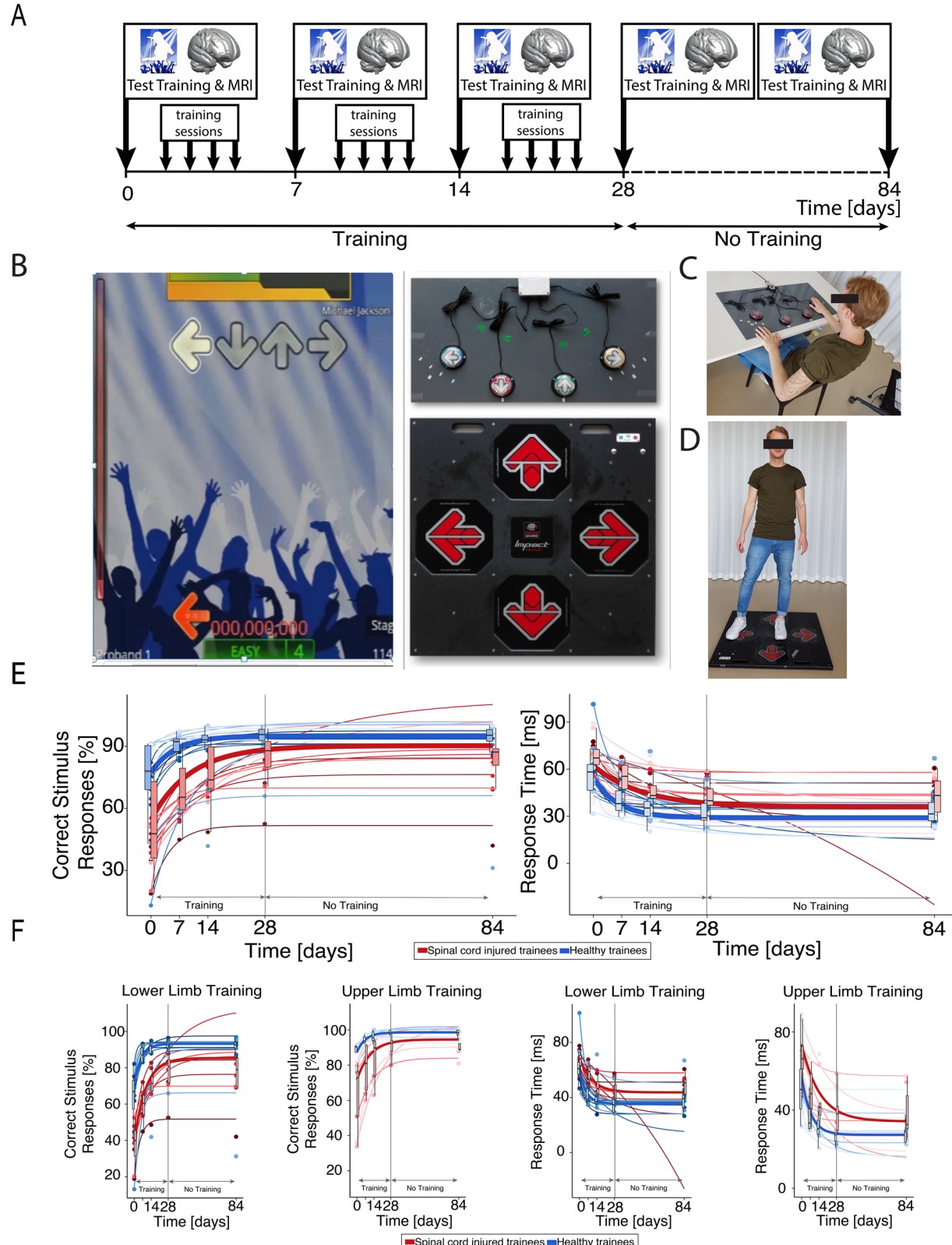

saturation (MTsat), and axial diffusivity (AD), and elevated mean diffusivity (MD) and radial diffusivity (RD). These bilateral differences affected key motor-related regions: the cerebellum, corticospinal tract (CST), thalamus, and cerebellar WM (Supplementary Table 2 and Supplementary Fig. 1). Compared to healthy non-trainees, SCI trainees showed greater linear increases in GM volume in the right primary motor cortex (M1) and in WM

volume within the cranial CST (corona radiata and pons) (Table 2 and Fig. 2). These deficits were accompanied by more negative quadratic WM volume changes in the same CST regions and bilateral cerebellum (Table 2 and Fig. 2). SCI trainees also showed larger positive quadratic MTsat changes in cerebellar GM and greater positive quadratic longitudinal relaxation rate ($R$1) changes in the left CST at the pons (Table 2). WM

**Fig. 1 | Experimental design, motor training task and longitudinal behavioral performance.** The experimental design (**A**) included magnetic resonance imaging (MRI) acquisition and training assessments at baseline (day 0), during the training period (days 7, 14, and 28), and at final retention assessment (day 84). Sixty minutes of supervised training in a motor-skill task was undertaken four times per week for 4 consecutive weeks (**B**), whereby participants were required to activate inputs with their hands or feet (depending on whether they were allocated to the upper (**C**) or lower limb (**D**) training groups) in response to rhythmic aural and visual stimuli in the dance game stepmania. The participant was tasked with selecting and activating the correct symbol at the precise moment the scrolling arrow overlapped with a set of static arrows at the top of the screen. Behavioral improvement, defined as the percentage of correct stimulus responses (%CSR), and response time (RT; the deviation in ms from the ideal response) were measured during a formal, standardized training assessment at weekly intervals (see methods). Spinal cord injured and healthy trainees values for these metrics are plotted as dots (blue: healthy trainees; red: spinal cord injured trainees; **E**: $n = 35$ and **F**: upper limb $n = 18$, lower limb $n = 17$), while participant-specific behavioral curves were computed (thin lines) along with the group median (thick line). Note that the results from the healthy controls trainees (blue lines) have been reported previously (Azzarito et al.[17]) and are shown for illustrative purposes. **B–D** Are adapted from Azzarito et al., NeuroImage (2023)[17], under a Creative Commons Attribution 4.0 International License.

microstructural changes accompanied these volumetric changes: SCI trainees had greater linear increases in FA and AD and linear RD decreases in the bilateral CST (corona radiata), as well as stronger negative quadratic FA and AD changes, and positive quadratic RD changes in the same regions (Table 2 and Fig. 2). Additionally, cerebellar WM showed more pronounced negative quadratic FA changes in SCI trainees (Table 2). Between the end of training (day 28) and follow-up (day 84), no significant changes were observed in SCI trainees across key brain regions (all $p > 0.05$; $BF_{01} > 1$; Table 2 and Supplementary Table 3), indicating stable structural changes post-training.

### Neuroplasticity and its association with performance improvements

We next investigated how the structural changes were related to performance improvements. In SCI trainees, greater linear GM volume increases in the right sensorimotor cortex larger RT improvements ($\delta$), while more negative quadratic changes in the right CST (internal capsule) were linked to faster RT gains ($\gamma$) and higher %CSR plateau ($\alpha$) (Table 3 and Fig. 3). Higher linear MTsat increases in the right cerebellar GM correlated with better RT improvement ($\delta$). Negative quadratic FA changes in left CST were linked to faster %CSR gains ($\gamma$). Both linear and negative quadratic FA changes in bilateral CST and left cerebellar WM were associated with higher %CSR improvement ($\delta$) and plateau ($\alpha$) (Table 3 and Fig. 3). Negative quadratic AD changes in left CST were associated with faster RT gains ($\gamma$). Greater linear AD increases in bilateral CST were associated with higher %CSR plateau ($\alpha$) and improvement ($\delta$). Similar associations were observed for negative quadratic AD in right CST (Table 3). In the left sensory cortex, negative quadratic MD changes were linked to shorter RT and a higher %CSR plateau ($\alpha$); less negative linear MD changes were linked to faster ($\gamma$) and larger ($\delta$) %CSR gains. Finally, greater quadratic RD changes in bilateral CST and negative linear RD changes in the right CST were linked to higher %CSR plateau ($\alpha$) and improvement ($\delta$) (Table 3 and Fig. 3).

### Somatotopic neuroplasticity differences between bimanual and bipedal training

We then assessed whether lower- and upper-limb trainees showed somatotopic changes. SCI lower-limb trainees showed greater positive quadratic MTsat changes in the left leg area of the sensorimotor cortex, while upper-limb trainees showed stronger linear MD decreases in the hand area (Table 4 and Fig. 4). SCI upper-limb trainees also had greater linear MTsat increases in the left thalamus. In the CST, SCI upper-limb trainees exhibited larger linear MTsat and WM volume changes at the pons and corona radiata. SCI lower-limb trainees showed greater linear volume increases and more pronounced positive quadratic $R1$ and negative quadratic WM volume changes in superior/rostral CST segments. They also had greater linear FA and AD increases in more posterior CST regions and stronger quadratic RD changes in leg-representing CST areas (Table 4 and Fig. 4). In the right cerebellum, SCI upper-limb trainees had larger linear WM volume increases and more pronounced positive quadratic $R1$ and negative quadratic volume changes. Contralaterally, they also showed larger linear MTsat and volume increases and stronger negative quadratic WM volume changes. In the left cerebellar WM, SCI upper-limb trainees had greater linear FA increases.

Finally, we assessed whether the plasticity response to training in SCI patients was similar to those observed in healthy trainees. Comparing SCI and healthy trainees, no significant differences were found in the linear ($\beta_1$) or quadratic ($\beta_2$) components of training-induced brain changes. Bayesian analysis supported the null hypothesis ($H_0$) across regions ($p > 0.12$; $BF_{01} > 1$; Fig. 2 and Table 2), indicating similar patterns of neuroplasticity in both groups. The only exception was in the right CST (corona radiata), where SCI trainees had a slightly smaller quadratic RD change than healthy trainees ($4.77 \pm 0.19$ vs. $4.97 \pm 0.23 \times 10^{-4}$ mm²/s/day²; Table 2).

### Discussion

In this longitudinal cohort of chronic SCI patients, intensive motor-skill training induced robust structural neuroplasticity across key motor-related brain regions, despite extensive prior injury. Structural changes emerged in bilateral sensorimotor cortices, corticospinal tracts, cerebellum, thalamus, and hippocampal formation—components of the cortico-cerebellar-hippocampal network underlying motor learning and memory. These micro- and macrostructural changes included both transient (initially increasing, then partially reverting) and persistent (progressive) components. The observed pattern—early expansions followed by partial renormalization, alongside gradual increases—likely reflects a sequence of biological processes during learning. Transient increases in GM volume and decreases in myelin-sensitive markers ($R1$, MTsat) observed early in training could correspond to synaptic proliferation and dendritic branching[5,19], wherein new connections temporarily inflate tissue volume and dilute myelin density. The subsequent plateau or reversal of these metrics may denote activity-dependent synaptic pruning and consolidation of circuits as the skill is mastered[20,21]. Meanwhile, the sustained increases in markers of myelination ($R1$, MTsat) and anisotropy (FA, AD) with continued practice are consistent with oligodendroglial adaptation and axonal remodeling that enhance fiber integrity and conduction efficiency[19]. The concomitant sensitivity of effective transverse relaxation rate ($R2^*$, a proxy for tissue iron content[22]) to training further supports the involvement of metabolic and glial changes, since iron is utilized during oligodendrocyte activity and neurovascular remodeling[14,23]. Although these mechanistic links are inferred from imaging and should be interpreted cautiously, their concordance with known cellular processes supports their biological plausibility[14].

Our findings align with prior neuroimaging studies in healthy individuals[3,4,16,17] and other patient populations demonstrating training-induced structural changes in both GM and WM[5–7]. Complex motor tasks (e.g., juggling or balance training) in healthy adults elicit volume increases in motor cortices and cerebellum and microstructural alterations in relevant WM pathways[4,24]. Similarly, rehabilitative training in patients with other conditions has been associated with structural brain changes in regions supporting the practiced functions[25]. Learning-related WM plasticity—particularly increased myelination and reorganized fiber architecture—has been observed in both humans and animal models, providing a mechanistic basis for the axonal and myelin-sensitive changes seen here.

At baseline, healthy individuals[17] as well as SCI upper-limb trainees showed a greater %CSR compared to their lower-limb counterparts. We argue that because upper limbs generally have higher dexterity than lower

**Table 2 | Training-induced neuroplasticity assessed using MRI during bimanual-bipedal motor learning**

| ROI | Map | Model parameter | Contrast | p-value (FEW-corrected) | Cluster size [voxels] | z-value | x [mm] | y [mm] | z [mm] | Fitted parameter SCI vs. healthy trainees | | | | Mean value in the cluster SCI trainees; 28 vs. 84 days | | | | Mean value in the cluster SCI vs. healthy trainees; day 84 | | |
|---|---|---|---|---|---|---|---|---|---|---|---|---|---|---|---|---|---|---|---|---|
| | | | | | | | | | | SCI | HC | p-value | BF$_{01}$ | SCI at day 28 | SCI at day 84 | p-value | BF$_{01}$ | HC at day 84 | p-value | BF$_{01}$ |
| GM L cerebellum | MT$_{sat}$ | $\beta_2$ | SCI trainees >non-trainees | 0.001 | 436 | 5.187 | −39 | −69 | −54 | 267.918 ± 1034.335 | −16.753 ± 64.388 | 0.274 | 1.818 | 1.892 ± 0.239 p.u. | 1.906 ± 0.175 p.u. | 0.620 | 3.495 | 1.989 ± 0.170 p.u. | 0.162 | 1.401 |
| GM R cerebellum | MT$_{sat}$ | $\beta_2$ | SCI trainees >non-trainees | <0.001 | 1311 | 5.755 | 27 | −79.5 | −52.5 | 312.139 ± 1083.225 | −10.738 ± 77.424 | 0.238 | 1.663 | 1.907 ± 0.217 p.u. | 1.884 ± 0.181 p.u. | 0.916 | 3.895 | 1.907 ± 0.116 p.u. | 0.663 | 2.846 |
| Corticospinal tract L at the level of the pons | R1 | $\beta_2$ | SCI trainees >non-trainees | 0.008 | 463 | 4.591 | −12 | −22.5 | −33 | 55.418 ± 276.519 | 3.267 ± 31.266 | 0.451 | 2.404 | 0.989 ± 0.061 s$^{-1}$ | 0.994 ± 0.087 s$^{-1}$ | 0.863 | 3.862 | 1.031 ± 0.078 s$^{-1}$ | 0.199 | 1.577 |
| R M1 | VBM | $\beta_1$ | SCI trainees >non-trainees | 0.021 | 195 | 6.927 | 27 | −22.5 | 45 | 0.402 ± 1.084 | −0.006 ± 0.092 | 0.142 | 1.168 | 0.065 ± 0.042 a.u. | 0.056 ± 0.013 a.u. | 0.249 | 2.123 | 0.052 ± 0.011 a.u. | 0.431 | 2.394 |
| Corticospinal tract L at the level of the corona radiata | VBM | $\beta_2$ | Non-trainees >SCI trainees | <0.001 | 241 | >10 | −34.5 | −9 | 27 | −7.266 ± 23.540 | 0.228 ± 0.590 | 0.208 | 1.523 | 0.700 ± 0.083 a.u. | 0.710 ± 0.087 a.u. | 0.544 | 3.302 | 0.734 ± 0.084 a.u. | 0.023 | 0.360 |
| Corticospinal tract L at the level of the pons | VBM | $\beta_2$ | Non-trainees >SCI trainees | 0.001 | 145 | >10 | −19.5 | −10.5 | 39 | −7.575 ± 30.296 | 0.368 ± 0.694 | 0.296 | 1.904 | 0.720 ± 0.088 a.u. | 0.728 ± 0.089 a.u. | 0.452 | 3.015 | 0.777 ± 0.074 a.u. | 0.060 | 0.717 |
| WM L cerebellum | VBM | $\beta_2$ | Non-trainees >SCI trainees | 0.018 | 79 | >10 | −22.5 | −51 | −42 | −18.658 ± 91.106 | −0.093 ± 1.217 | 0.413 | 2.299 | 0.612 ± 0.110 a.u. | 0.618 ± 0.103 a.u. | 0.090 | 1.041 | 0.734 ± 0.084 a.u. | 0.001 | 0.028 |
| Corticospinal tract R at the level of the pons | VBM | $\beta_1$ | SCI trainees >non-trainees | 0.038 | 61 | >10 | 4.5 | −28.5 | −37.5 | 0.731 ± 1.851 | −0.003 ± 0.111 | 0.122 | 1.046 | 0.781 ± 0.117 a.u. | 0.786 ± 0.117 a.u. | 0.763 | 3.754 | 0.901 ± 0.113 a.u. | 0.006 | 0.129 |
| Corticospinal tract R at the level of the corona radiata | VBM | $\beta_2$ | Non-trainees >SCI trainees | <0.001 | 1586 | >10 | 25.5 | −13.5 | 40.5 | −9.642 ± 34.869 | 0.047 ± 0.297 | 0.269 | 1.796 | 0.702 ± 0.082 a.u. | 0.713 ± 0.087 a.u. | 0.369 | 2.698 | 0.777 ± 0.074 a.u. | 0.025 | 0.380 |
| WM R cerebellum | VBM | $\beta_2$ | Non-trainees >SCI trainees | 0.005 | 105 | 6.982 | 24 | −46.5 | −40.5 | −20.956 ± 99.259 | 0.613 ± 1.691 | 0.384 | 2.210 | 0.638 ± 0.099 a.u. | 0.637 ± 0.098 a.u. | 0.512 | 3.211 | 0.738 ± 0.097 a.u. | 0.004 | 0.102 |
| Corticospinal tract L at the level of the corona radiata | FA | $\beta_1$ | SCI trainees >non-trainees | <0.001 | 77 | >10 | −21 | −10.5 | 34.5 | 0.094 ± 0.238 | 0.002 ± 0.007 | 0.131 | 1.670 | 0.462 ± 0.047 | 0.470 ± 0.049 | 0.227 | 1.949 | 0.484 ± 0.044 | 0.377 | 3.013 |
| WM L cerebellum | FA | $\beta_2$ | Non-trainees >SCI trainees | <0.001 | 39 | 6.910 | −13.5 | −36 | −42 | −101.912 ± 383.704 | 2.543 ± 8.180 | 0.278 | 1.835 | 0.523 ± 0.035 | 0.528 ± 0.034 | 0.852 | 3.750 | 0.542 ± 0.005 | 0.156 | 1.338 |
| Corticospinal tract R at the level of the corona radiata | FA | $\beta_1$ | SCI trainees >non-trainees | <0.001 | 182 | >10 | 22.5 | −12 | 33 | 0.100 ± 0.412 | 0.000 ± 0.000 | 0.332 | 2.039 | 0.445 ± 0.059 | 0.448 ± 0.057 | 0.564 | 3.269 | 0.464 ± 0.009 | 0.358 | 2.170 |
| Corticospinal tract L at the level of the corona radiata | AD | $\beta_1$ | SCI trainees >non-trainees | <0.001 | 52 | >10 | −22.5 | −15 | 34.5 | $(1.98 ± 7.63) × 10^{-4}$ | $(-0.01 ± 0.04) × 10^{-4}$ | 0.298 | 1.912 | $(10.26 ± 0.53) × 10^{-4}$ mm²/s | $(10.15 ± 0.54) × 10^{-4}$ mm²/s | 0.307 | 2.357 | $(10.32 ± 0.17) × 10^{-4}$ mm²/s | 0.341 | 2.136 |
| Corticospinal tract R at the level of the corona radiata | AD | $\beta_1$ | SCI trainees >non-trainees | <0.001 | 182 | >10 | 21 | −19.5 | 46.5 | $(2.86 ± 11.80) × 10^{-4}$ | $(-0.00 ± 0.00) × 10^{-4}$ | 0.332 | 2.039 | $(10.32 ± 0.528) × 10^{-4}$ mm²/s | $(10.26 ± 0.48) × 10^{-4}$ mm²/s | 0.251 | 2.076 | $(10.31 ± 0.13) × 10^{-4}$ mm²/s | 0.800 | 2.998 |
| Sensorimotor cortices L | MD | $\beta_2$ | SCI trainees >non-trainees | 0.001 | 38 | 6.706 | −31.5 | −49.5 | 61.5 | $(879.94 ± 3288.14) × 10^{-4}$ | $(-12.07 ± 83.48) × 10^{-4}$ | 0.280 | 1.842 | $(9.40 ± 0.88) × 10^{-4}$ mm³/s | $(9.38 ± 1.16) × 10^{-4}$ mm²/s | 0.757 | 3.646 | $(8.83 ± 0.27) × 10^{-4}$ mm²/s | 0.170 | 1.445 |

**Table 2 (continued) | Training-induced neuroplasticity assessed using MRI during bimanual-bipedal motor learning**

| ROI | Map | Model parameter | Contrast | p-value (FEW-corrected) | Cluster size [voxels] | z-value | x [mm] | y [mm] | z [mm] | Fitted parameter SCI vs. healthy trainees | | p-value | BF$_{01}$ | Mean value in the cluster SCI trainees; 28 vs. 84 days | | p-value | BF$_{01}$ | Mean value in the cluster SCI vs. healthy trainees; day 84 | p-value | BF$_{01}$ |
|---|---|---|---|---|---|---|---|---|---|---|---|---|---|---|---|---|---|---|---|---|
| | | | | | | | | | | SCI | HC | | | SCI at day 28 | SCI at day 84 | | | HC at day 84 | | |
| Corticospinal tract L at the level of the corona radiata | RD | $\beta_2$ | SCI trainees > non-trainees | <0.001 | 54 | >10 | −21 | −15 | 36 | $(59.83 \pm 225.38) \times 10^{-4}$ | $(0.70 \pm 2.27) \times 10^{-4}$ | 0.295 | 1.903 | $(5.15 \pm 0.47) \times 10^{-4}$/s | $(5.09 \pm 1.00) \times 10^{-4}$ mm²/s | 0.416 | 2.809 | $(4.94 \pm 0.06) \times 10^{-4}$ mm²/s | 0.279 | 1.882 |
| Corticospinal tract R at the level of the corona radiata | RD | $\beta_2$ | SCI trainees > non-trainees | <0.001 | 190 | >10 | 30 | −12 | 36 | $(4.77 \pm 0.19) \times 10^{-4}$ | $(4.97 \pm 0.23) \times 10^{-4}$ | 0.010 | 2.039 | $(4.98 \pm 0.43) \times 10^{-4}$/s | $(4.77 \pm 0.19) \times 10^{-4}$ mm²/s | 0.321 | 3.126 | $(4.76 \pm 0.06) \times 10^{-4}$ mm²/s | 0.209 | 2.096 |

Longitudinal statistical parametric mapping (SPM) shows differences between spinal cord injury (SCI) trainees and healthy non-trainees in both the linear ($\beta_1$) and quadratic ($\beta_2$) components of the quadratic model describing the temporal evolution of MRI readouts. These include gray matter (GM) volume, white matter (WM) volume, myelin-sensitive longitudinal relaxation rate ($R1$), magnetization transfer rate (MTsat), fractional anisotropy (FA), mean diffusivity (MD), radial diffusivity (RD), and axial diffusivity (AD). Also included are post-hoc analyses comparing MRI readouts within the cluster: (i) in SCI trainees between days 28 and 84, and (ii) at day 84 between SCI and healthy trainees (HC). For each combination of MRI readout and model parameter ($\beta_1$ or $\beta_2$), only the largest cluster within a distinct brain region of interest (ROI) is shown. See Supplementary Table 3 for all significant clusters. R right, L left, p.u. percent units, a.u. arbitrary units.

limbs, participants using their upper limbs achieved better performance on the same baseline tasks. All patients achieved significant performance gains over the training period, and these functional improvements were accompanied by widespread changes in both GM and WM. To some degree, particularly at the beginning of the training period, the performance gains could also be attributed to familiarization processes. Importantly, the magnitude of these structural changes correlated with individual performance improvements in the game, indicating that these neuroplastic responses are functionally meaningful rather than epiphenomenal. Associations between structural changes and behavioral improvements were further supported by findings showing that faster myelin-related increases tracked with quicker reaction time improvements. These results suggest that neuroimaging may provide early indications of how well a patient is responding to therapy. In the future, such markers might help clinicians individualize training intensity or duration—e.g., by identifying patients who might benefit from extended training to reach normalization of certain brain metrics relative to healthy benchmarks.

Moreover, Bayesian analyses revealed little evidence for group differences between SCI patients and healthy controls in most structural measures, underscoring the preserved neuroplastic potential even in chronic SCI. This study offers the first comprehensive in vivo evidence that chronic SCI does not eliminate the brain's capacity for training-induced structural remodeling, extending prior findings in healthy adults to established CNS injury. In some cases, changes in SCI patients even exceeded those of controls, showing that severe impairments do not preclude meaningful plasticity. We further explored whether there are differences between the upper-limb SCI trainees and healthy upper-limb trainees, as well as between the lower-limb SCI trainees and healthy lower-limb trainees. We found evidence for greater responses in the SCI patients (see Supplementary Table 6); however, changes in the opposite direction were also observed, potentially reflecting different adaptations in mastering the training task between SCI patients and healthy trainees. Nevertheless, further investigations are needed to clarify whether the underlying biological processes causing the observed alterations in $R1$, MTsat, $R2^*$, FA, MD, AD, or RD are identical to those in healthy controls, or whether additional or more pronounced biological processes are responsible for these findings. It should be noted that the participating SCI patients had been living with their injury for an average of 95 months, and the observed neuroplastic changes followed a similar pattern to those reported in healthy controls. Specifically, an initial decrease (~0–14 days) followed by an increase (~14–28 days) in MT, $R1$, and $R2^*$ has been described in healthy individuals[17], along with linear increases in FA and AD and decreases in RD and MD[16]. These patterns were also observed in SCI patients, suggesting that pathological processes, typically described as mainly linear and occurring in the opposite direction[1,26], may not have played a substantial role in the present findings. Whether the initial and subsequent phases should be viewed as one continuous process or as two distinct and independent processes requires further investigation.

The anatomical specificity of the neuroplastic changes provides insight for tailoring rehabilitation. We found evidence of somatotopic differentiation in arm-associated regions of the CST and sensorimotor cortex compared to lower-limb training[27], similar to what has been reported in healthy controls when upper- and lower-limb training was compared[16,17]. This specificity suggests that training regimens engaging the affected limb or task-specific circuits may yield the greatest neuroplastic benefits, informing personalized therapy plans. We observed that many training-related changes in SCI patients occurred in brain regions with relatively preserved baseline tissue integrity, consistent with previous observations that neuroplasticity tends to be most pronounced in less degenerated areas[1]. Furthermore, there are some indications of a higher prevalence of significant clusters of linear or quadratic volumetric or microstructural changes in the left hemisphere, which could be related to the higher number of right-handed participants. However, whether this asymmetry is exclusively due to handedness, preferential use of the dominant hand, or other factors still needs to be determined. Nevertheless, we also detected plastic changes, albeit smaller, in some regions that had shown atrophy or demyelination due to

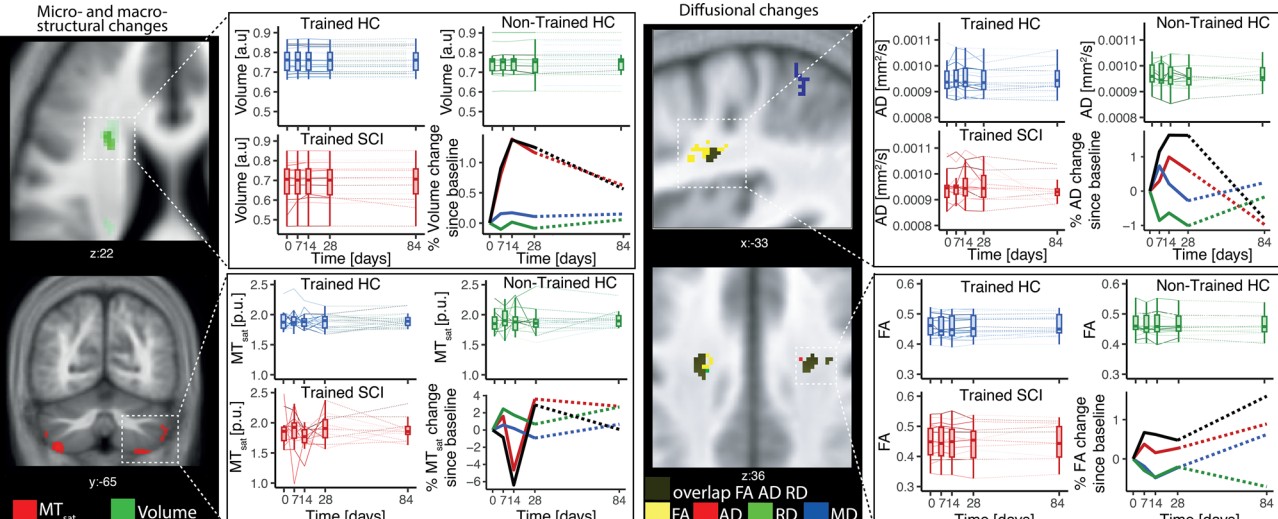

**Fig. 2 | Training-induced neuroplasticity assessed using MRI during bimanual-bipedal motor learning.** Training-induced changes during the learning of the motor skill task, combining all spinal cord injured (SCI) upper and lower limb trainees (in red, n = 17), compared to healthy control trainees (in blue, n = 18) and healthy non-trainees (in green, n = 14). The black line indicates the differences between SCI trainees and non-trainees (SCI trainees–Non-trainees). Dashed lines represent periods without training. On the left column, micro-macrostructural changes are depicted with myelin-sensitive magnetization transfer (*MTsat*, in red), longitudinal relaxation rate (*R1*, in yellow), and volume metric changes (in green). On the right column, diffusional changes are illustrated with fractional anisotropy (*FA*, in yellow), mean diffusivity (*MD*, in blue), axial diffusivity (*AD*, in red), and radial diffusivity (*RD*, in green). The bold blue and green lines represent results from healthy trainees, reported previously (Azzarito et al.[17]; Emmenegger et al.[16]) and are shown for illustrative purposes.

the injury (Supplementary Fig. 2), a phenomenon that has also been hinted at in stroke recovery and other neurological conditions[6,25,28]. The observed structural remodeling within chronically affected circuits suggests that rehabilitative training can engage adaptive mechanisms beyond fully spared pathways. Nevertheless, it should be considered that further investigations focusing on these cases are needed to determine whether different biological processes occur in these brain regions compared to others. This is particularly important since the SCI trainees were compared to healthy controls rather than to a non-training chronic SCI group, making it difficult to draw definitive conclusions about the specific biological processes within these regions. However, it could be assumed that equal or even greater differences in qMRI metrics might be observed, as the known pathological processes in SCI (e.g., axonal degeneration and demyelination)[1,26] would counteract the suggested biological processes, such as axonal sprouting or increased myelination[5].

The persistence of structural changes at the 2-month follow-up (in the absence of further training) implies that relatively short, intensive rehabilitation—consisting of approximately 30 min of active task performance within a 60-min training session—can induce lasting neural reorganization. This durability is encouraging for patients and clinicians, as it indicates that gains achieved during a focused training period might be maintained, reducing the risk of rapid regression after therapy cessation.

These findings have important clinical implications for SCI rehabilitation. They demonstrate that chronic SCI patients retain substantial capacity for adaptive brain reorganization, which can be harnessed with appropriate strategies. Similar trajectories in patients and controls suggest that, with intensive practice, the injured brain can re-engage learning processes seen in healthy individuals. This supports the inclusion of challenging, skill-based training even long after injury, as it can promote structural brain changes and functional gains despite ongoing neurodegeneration. Engaging, cognitively and physically demanding tasks—like dance-based games—may boost compliance and amplify effects. Combined with adjunct therapies, such training could enhance neurological function and quality of life in SCI.

Our study has limitations. The cohort was small (17 male patients) and limited to patients in the chronic phase, reducing generalizability to females, acute injuries, or other severities. While this restriction may limit the

generalizability of the findings to female populations, the higher prevalence of traumatic SCI in men[29] supports the relevance of the study to the majority of affected patients. Although we included a non-training control group and standardized scan timing, unmonitored factors like sleep or hydration may have influenced results. Furthermore, medication use was not assessed or controlled for and may have influenced the observed differences between patient subgroups as well as comparisons with healthy controls. The imaging metrics, while informative, are indirect proxies of microstructure and lack cellular specificity; histological validation would strengthen our interpretations. Moreover, we did not assess whether gains in game-based performance translated to real-world motor function. Longer training and follow-up are needed to determine whether neuroplastic changes can be enhanced or sustained over time. A further issue to consider is that the SCI upper- and lower-limb trainees showed significant differences in lesion level, with upper-limb trainees having higher lesion levels compared to lower-limb trainees. This group allocation was intentional to achieve comparable motor capacity in both groups for their respective training tasks. This baseline difference could nevertheless have influenced behavioral performance, as a higher %CSR was found in SCI upper-limb trainees compared to SCI lower-limb trainees at baseline, without a significant difference in RT. We note, however, that this baseline difference was also observed in healthy individuals[17]. Furthermore, it cannot be excluded that differences in lesion level may have concealed some of the somatotopic neuroplasticity differences, as higher lesion levels have been shown to lead to greater neurodegeneration compared to lower lesion levels[1,30,31], potentially counteracting neuroplastic changes to varying degrees. Furthermore, learning was modeled using an exponential fit, which may not always be the most appropriate model for every individual (Fig. 1E, F). Therefore, to avoid bias from such cases, behavioral outliers were excluded from the statistical analyses. This applied, for example, to a patient in the lower-limb training group for whom an exponential fit produced unphysical values for RT (see Fig. 1E, F).

In summary, these findings challenge the notion that chronic CNS injury irreversibly impairs the brain's adaptive capacity. They show that plasticity-related mechanisms remain accessible well after injury and point toward generalizable principles of rehabilitation that may apply across neurological disorders. The application of quantitative, biologically sensitive

**Table 3 | Association between MRI-derived neuroplastic changes and bimanual-bipedal motor learning performance**

| ROI | MRI model parameter | Behavioral improvement | Direction of association | $p_{FWE}$ value | Cluster size [voxels] | z-value | x [mm] | y [mm] | z [mm] |
|---|---|---|---|---|---|---|---|---|---|
| Cerebellum GM R | $\beta_1$ for MTsat | RT improvement (δ) | Negative | 0.002 | 462 | 4.030 | 12 | −73.5 | −13.5 |
| Sensorimotor cortex R | $\beta_2$ for volume | RT improvement (δ) | Positive | 0.032 | 151 | 4.255 | 39 | −36 | 33 |
| CST at the level of the capsule internal R | $\beta_2$ for volume | RT improvement (δ) | Negative | 0.008 | 101 | 5.805 | 18 | −3 | 10.5 |
| CST at the level of the corona radiata R | $\beta_2$ for volume | %CSR plateauing (α) | Negative | <0.001 | 245 | 3.735 | 22.5 | −27 | 30 |
| CST at the level of the corona radiata L | $\beta_1$ for FA | %CSR improvement speed (γ) | Positive | <0.001 | 32 | 5.266 | −27 | −4.5 | 21 |
| CST at the level of the corona radiata L | $\beta_1$ for FA | %CSR plateauing (α) | Positive | <0.001 | 553 | 4.397 | −15 | −10.5 | 36 |
| Cerebellum WM L | $\beta_2$ for FA | %CSR plateauing (α) | Negative | <0.001 | 210 | 5.436 | −13.5 | −33 | −27 |
| CST at the level of the corona radiata L | $\beta_1$ for FA | %CSR improvement (δ) | Positive | <0.001 | 31 | 4.336 | −31.5 | −6 | 33 |
| Cerebellum WM L | $\beta_2$ for FA | %CSR improvement (δ) | Negative | <0.001 | 60 | 4.511 | −19.5 | −49.5 | −33 |
| CST at the level of the corona radiata R | $\beta_1$ for FA | %CSR plateauing (α) | Positive | <0.001 | 504 | 4.346 | 25.5 | −24 | 40.5 |
| CST at the level of the corona radiata R | $\beta_2$ for FA | %CSR improvement (δ) | Negative | <0.001 | 176 | 5.047 | 28.5 | 12 | 36 |
| CST at the level of the corona radiata R | $\beta_1$ for FA | %CSR plateauing (α) | Positive | <0.001 | 42 | 4.808 | 19.5 | 4.5 | 45 |
| CST at the level of the corona radiata L | $\beta_1$ for AD | %CSR improvement speed (γ) | Positive | <0.001 | 31 | 5.262 | −27 | −6 | 22.5 |
| CST at the level of the corona radiata L | $\beta_1$ for AD | %CSR plateauing (α) | Positive | <0.001 | 513 | 4.294 | −30 | −4.5 | 39 |
| CST at the level of the corona radiata L | $\beta_1$ for AD | %CSR improvement (δ) | Positive | <0.001 | 29 | 4.342 | −28.5 | −1.5 | 37.5 |
| CST at the level of the crus cerebri R | $\beta_2$ for AD | RT improvement (δ) | Positive | 0.001 | 25 | 3.616 | 9 | −25.5 | −13.5 |
| CST at the level of the capsula interna R | $\beta_1$ for AD | %CSR plateauing (α) | Positive | <0.001 | 188 | 4.678 | 16.5 | −7.5 | 15 |
| CST at the level of the corona radiata R | $\beta_1$ for AD | %CSR plateauing (α) | Positive | <0.001 | 490 | 4.351 | 21 | −24 | 43.5 |
| CST at the level of the crus cerebri R | $\beta_1$ for AD | %CSR plateauing (α) | Positive | <0.001 | 29 | 4.322 | 9 | −9 | −13.5 |
| CST at the level of the corona radiata R | $\beta_2$ for AD | %CSR plateauing (α) | Negative | <0.001 | 159 | 5.195 | 27 | 16.5 | 28.5 |
| CST at the level of the crus cerebri R | $\beta_2$ for AD | %CSR plateauing (α) | Negative | <0.001 | 32 | 4.066 | 18 | −13.5 | −4.5 |
| CST at the level of the pons R | $\beta_2$ for AD | %CSR plateauing (α) | Negative | <0.001 | 238 | 4.061 | 3 | −27 | −45 |
| CST at the level of the corona radiata R | $\beta_1$ for AD | %CSR improvement (δ) | Positive | <0.001 | 39 | 4.917 | 13.5 | −9 | 31.5 |
| Sensorimotor cortex L | $\beta_2$ for MD | %CSR plateauing (α) | Positive | <0.001 | 171 | 4.601 | −49.5 | −7.5 | 43.5 |
| Sensorimotor cortex L | $\beta_1$ for MD | %CSR improvement (δ) | Negative | 0.028 | 20 | 4.602 | −3 | −25.5 | 73.5 |
| CST at the level of the corona radiata L | $\beta_2$ for RD | %CSR plateauing (α) | Positive | <0.001 | 170 | 3.970 | −31.5 | −7.5 | 30 |
| CST at the level of the corona radiata R | $\beta_1$ for RD | %CSR plateauing (α) | Negative | <0.001 | 201 | 4.736 | 27 | 15 | 28.5 |
| CST at the level of the corona radiata R | $\beta_2$ for RD | %CSR plateauing (α) | Positive | <0.001 | 446 | 4.152 | 24 | −21 | 37.5 |

Correlations between the linear ($\beta_1$) and quadratic ($\beta_2$) components of the quadratic model describing the temporal evolution of MRI readouts and the performance improvement. MRI readouts include magnetization transfer rate (MTsat), volume changes (VBM), fractional anisotropy (FA), mean diffusivity (MD), radial diffusivity (RD), axial diffusivity (AD) maps and response time (RT) and percentage of correct stimulus responses (CSR%). For each combination of MRI readout, model parameter ($\beta_1$ or $\beta_2$), and performance improvement, only the largest cluster within a distinct brain region of interest (ROI) is shown. See Supplementary Table 4 for all significant clusters. R right, L left.

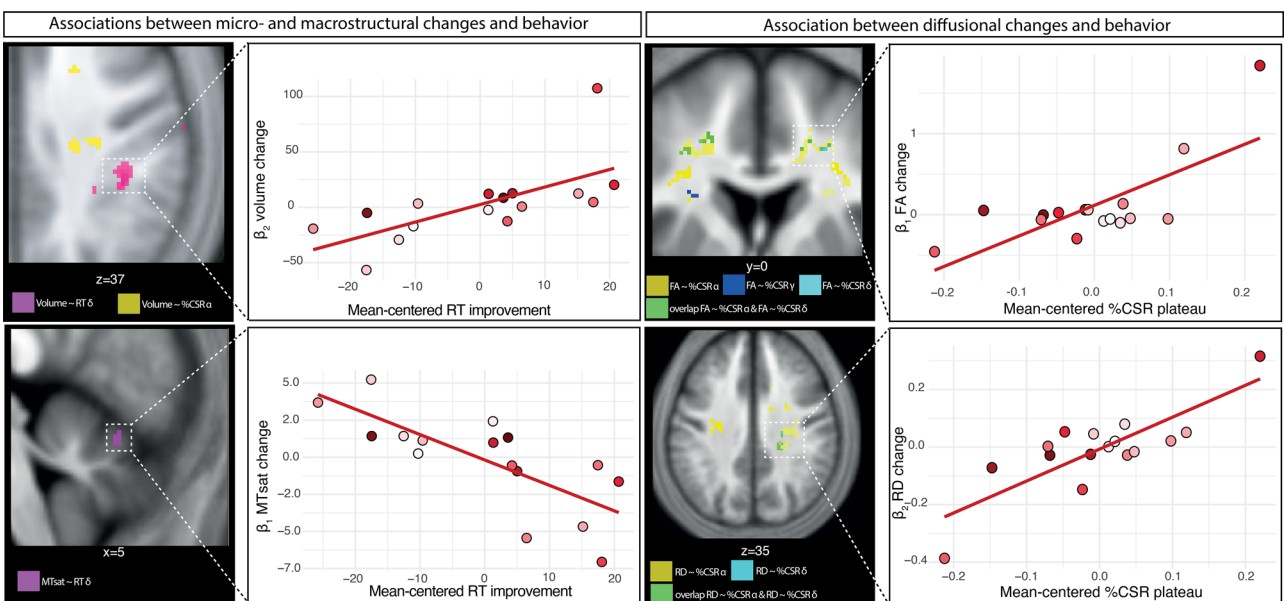

**Fig. 3 | Association between MRI-derived neuroplastic changes and bimanual-bipedal motor learning performance.** Associations between linear ($\beta_1$) or quadratic ($\beta_2$) micro- and macrostructural changes (left column: MTsat = magnetization transfer saturation and volume) or diffusion parameters (right column: FA = fractional anisotropy, RD = radial diffusivity) and performance improvements parameters (response time = RT, percentage of correct stimulus responses = %CSR, $\alpha$ = plateau, $\delta$ = improvement, and $\gamma$ = improvement speed) of spinal cord injury trainees. $n = 16$.

qMRI allows us to track coordinated, circuit-level adaptations—offering a framework for future mechanistic studies and targeted therapeutic strategies.

## Methods
### Participants
This study included 32 healthy males (23–62 years) and 17 males with chronic SCI (23–70 years, >6 months post-injury; Table 1). All healthy participants and most patients were right-handed, while four SCI participants showed a preference for the left hand, as assessed with the Edinburgh Handedness Inventory[32], but this was likely affected by post-injury preferences on account of asymmetrical limb function. Recruitment was limited to males to avoid potential sex-related confounding effects on neuroplastic processes, which may differ in temporal dynamics, spatial distribution, or magnitude. Participants provided written informed consent. The study complied with the Declaration of Helsinki and Good Clinical Practice guidelines and was approved by the Zurich Cantonal Ethics Committee (KEK-2013-0559). All ethical regulations relevant to human research participants were followed.

Participants underwent longitudinal MRI and behavioral assessments at baseline (day 0), during training (days 7, 14, 28), and at follow-up (day 84; Fig. 1A). Healthy non-trainees followed the same MRI schedule without training. On assessment days, MRI was performed before any tasks, and participants were instructed to maintain normal routines and avoid new physical or dance activities. Participants were divided into five groups: healthy upper-limb trainees ($n = 9$), healthy lower-limb trainees ($n = 9$), non-trainees ($n = 14$), SCI upper-limb trainees ($n = 9$), and SCI lower-limb trainees ($n = 8$). Grouping was based on motor capacity; patients without residual lower-limb function were excluded from lower-limb training. All participants had normal or corrected vision, no psychiatric or neurological conditions (aside from SCI), and no MRI contraindications. There were no significant age differences across groups (ANOVA: $F = 1.55$, $p = 0.205$): healthy upper limb ($34.48 \pm 10.94$ years), healthy lower limb ($39.26 \pm 13.98$), non-trainees ($38.89 \pm 11.18$), SCI upper limb ($46.46 \pm 18.94$), and SCI lower limb ($47.60 \pm 10.04$). Among SCI trainees, American Spinal Injury Association Impairment Scale (AIS) grades (A–D) were similarly distributed across upper and lower limb subgroups (Fisher's

exact test: $p = 0.390$), as were injury completeness (Fisher's exact test: $p = 0.294$) and time since injury (Welch's t-test: $p = 0.774$, Table 1). However, the SCI trainees differed significantly regarding lesion level (Welch's t-test: $p = 0.008$), with more tetraplegic participants in the upper limb group compared to the lower limb subgroup (Table 1). None had prior dance experience or exposure to the experimental setup. Neuroplastic changes in the healthy cohort have been described previously, showing structural adaptations in the motor, hippocampal, and cerebellar system[16,17].

### Motor skill learning
Training lasted 4 weeks, with 60-min sessions (including breaks) four times per week (Fig. 1A). No practice occurred between day 28 and the follow-up on day 84. StepMania 5 (beta 3) was used for training, with custom input devices based on the trained limb. Lower-limb training used a dance mat (impact dance platform) connected to a monitor; SCI patients unable to stand wore a ceiling-mounted harness. Upper-limb training used a tabletop interface; participants responded with left and right hands to directional cues on a laptop. Each session consisted of 15 bouts, each lasting 2 min, totaling 30 minutes of training, with roughly 2-min breaks between bouts. During each bout, arrows scrolled to music, and participants responded by triggering inputs when the arrows reached a fixed target. Arrow sequences were generated with the "dancing monkeys" script (https://monket.net/dancing-monkeys/) to vary complexity while ensuring feasibility. Scoring awarded 2 points for responses within 45 ms, 1 point for 45–90 ms, and 0 for >90 ms delays. Feedback was provided as a percentage of the total possible score. Difficulty increased with arrow density, speed, and complexity. Advancement required ≥80% in three nonconsecutive bouts; failure in three consecutive bouts <30% triggered demotion. The total training time was designed to be equal for the left and right sides; however, participants were free to compensate by using their stronger side for most inputs. Although this was not explicitly measured, it was likely to be the case in some instances of highly unilateral injuries. Standardized assessments were conducted at baseline, days 7, 14, 28, and 84 using a 3:20-min sequence of novel arrow patterns with tempos of 60, 80, 100, and 120 BPM. Backgrounds were black and accompanied by a metronome, removing musical cues. Directional cues were balanced to avoid bias. Primary metrics were percent correct stimulus responses (%CSR) and the response time (RT). %CSR captured responses

**Table 4 | Somatotopic differences associated with upper versus lower limbs training**

| ROI | Map | Model parameter | Contrast | $p_{FWE}$ value | Cluster size [voxels] | z-value | x [mm] | y [mm] | z [mm] |
|---|---|---|---|---|---|---|---|---|---|
| Sensorimotor cortices L | $MT_{sat}$ | $\beta_2$ | SCI trainees upper < lower | 0.029 | 219 | 4.304 | −9 | −46.5 | 76.5 |
| Thalamus L | $MT_{sat}$ | $\beta_1$ | SCI trainees upper > lower | 0.005 | 346 | 4.111 | 4.5 | −15 | 10.5 |
| Corticospinal tract at the level of the pons L | $MT_{sat}$ | $\beta_1$ | SCI trainees upper > lower | 0.001 | 365 | 4.583 | −4.5 | −30 | −33 |
| Cerebellum WM L | $MT_{sat}$ | $\beta_1$ | SCI trainees upper > lower | 0.006 | 248 | 4.482 | −21 | −49.5 | −39 |
| Corticospinal tract at the level of the pons R | $MT_{sat}$ | $\beta_1$ | SCI trainees upper > lower | <0.001 | 685 | 4.518 | 12 | −33 | −33 |
| Corticospinal tract at the level of the pons L | $R1$ | $\beta_2$ | SCI trainees upper > lower | 0.044 | 219 | 3.994 | −1.5 | −21 | −27 |
| Cerebellum WM R | $R1$ | $\beta_2$ | SCI trainees upper < lower | 0.029 | 303 | 3.558 | 28.5 | −55.5 | −40.5 |
| Corticospinal tract at the level of the corona radiata L | VBM | $\beta_1$ | SCI trainees upper < lower | <0.001 | 234 | >10 | −24 | −7.5 | 36 |
| Corticospinal tract at the level of the pons L | VBM | $\beta_1$ | SCI trainees upper > lower | <0.001 | 203 | >10 | −4.5 | −30 | −33 |
| Cerebellum WM L | VBM | $\beta_1$ | SCI trainees upper > lower | <0.001 | 210 | >10 | −22.5 | −51 | −40.5 |
| Corticospinal tract at the level of the corona radiata L | VBM | $\beta_2$ | SCI trainees upper < lower | <0.001 | 252 | >10 | −24 | −7.5 | 36 |
| Corticospinal tract at the level of the pons L | VBM | $\beta_2$ | SCI trainees upper < lower | 0.001 | 162 | >10 | −4.5 | −30 | −33 |
| Cerebellum WM L | VBM | $\beta_2$ | SCI trainees upper < lower | 0.029 | 68 | >10 | −22.5 | −51 | −40.5 |
| Corticospinal tract at the level of the corona radiata R | VBM | $\beta_1$ | SCI trainees upper < lower | <0.001 | 1470 | >10 | 25.5 | −13.5 | 40.5 |
| Cerebellum WM R | VBM | $\beta_1$ | SCI trainees upper > lower | <0.001 | 197 | >10 | 22.5 | −46.5 | −40.5 |
| Corticospinal tract at the level of the pons R | VBM | $\beta_2$ | SCI trainees upper < lower | 0.036 | 62 | 4.371 | 6 | −27 | −15 |
| Corticospinal tract at the level of the corona radiata R | VBM | $\beta_2$ | SCI trainees upper < lower | <0.001 | 1430 | >10 | 25.5 | −13.5 | 40.5 |
| Cerebellum WM R | VBM | $\beta_2$ | SCI trainees upper > lower | 0.008 | 95 | 6.918 | 24 | −46.5 | −40.5 |
| Corticospinal tract at the level of the corona radiata L | FA | $\beta_1$ | SCI trainees upper < lower | <0.001 | 109 | >10 | −33 | −9 | 27 |
| Corticospinal tract at the level of the corona radiata L | FA | $\beta_1$ | SCI trainees upper < lower | 0.003 | 26 | 6.914 | −4.5 | −27 | −30 |
| Cerebellar peduncle L | FA | $\beta_1$ | SCI trainees upper > lower | 0.011 | 20 | 6.623 | −12 | −42 | −37.5 |
| Corticospinal tract at the level of the corona radiata R | FA | $\beta_1$ | SCI trainees upper < lower | <0.001 | 182 | >10 | 22.5 | −13.5 | 33 |
| Corticospinal tract at the level of the corona radiata L | AD | $\beta_1$ | SCI trainees upper < lower | 0.002 | 25 | >10 | −36 | −9 | 27 |
| Corticospinal tract at the level of the corona radiata R | AD | $\beta_1$ | SCI trainees upper < lower | <0.001 | 182 | >10 | 22.5 | −18 | 46.5 |
| Sensorimotor cortices L | MD | $\beta_1$ | SCI trainees upper < lower | <0.001 | 45 | 6.134 | −31.5 | −46.5 | 60 |
| Corticospinal tract at the level of the corona radiata L | RD | $\beta_2$ | SCI trainees upper < lower | <0.001 | 54 | >10 | −21 | −15 | 36 |
| Corticospinal tract at the level of the corona radiata R | RD | $\beta_2$ | SCI trainees upper < lower | <0.001 | 191 | >10 | 30 | −12 | 36 |

Longitudinal statistical parametric mapping (SPM) shows differences between upper-limb and lower-limb spinal cord injury (SCI) trainees in both the linear ($\beta_1$) and quadratic ($\beta_2$) components of the quadratic model describing the temporal evolution of MRI readouts. These include longitudinal rate ($R1$) and magnetization transfer rate (MTsat), transverse relaxation rate ($R2^*$), fractional anisotropy (FA), mean diffusivity (MD), radial diffusivity (RD), and axial diffusivity (AD). For each combination of MRI readout and model parameter ($\beta_1$ or $\beta_2$), only the largest cluster within a distinct brain region of interest (ROI) is shown. See Supplementary Table 5 for all significant clusters R = right, L = left.

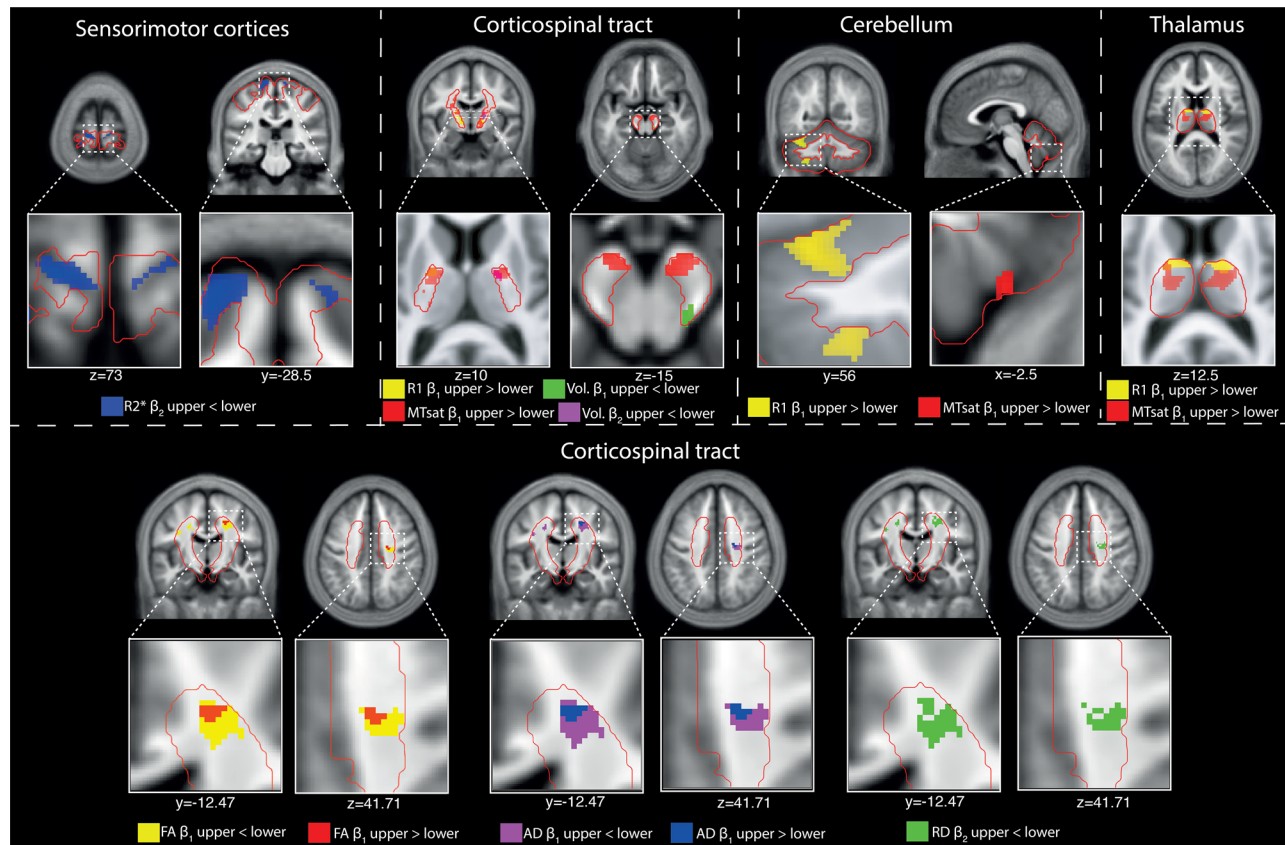

**Fig. 4 | Somatotopic differences associated with upper versus lower limbs training in SCI patients.** In spinal cord injury (SCI) patients, lower limb training resulted in a greater quadratic concave (u-shaped) change in transverse relaxation rate ($R2^*$; blue) in the bilateral lower limb area of the bilateral sensorimotor cortices. Training of the upper limbs resulted in greater linear changes in magnetization transfer saturation (MTsat; red), longitudinal relaxation rate ($R1$; yellow), and quadratic convex (n-shaped) white matter volume (vol. quad; magenta) changes along the corticospinal tract in upper limb areas. Note that $R1$ and MTsat clusters overlap in the left posterior limb of the internal capsule, while in the right posterior limb of the internal capsule, the quadratic volume change cluster overlaps with the cluster of linear MTsat changes. Further, in the right cerebral peduncle, SCI lower-limb trainees showed a greater linear increase in the lower limb area compared to upper-limb trainees. Training of the upper limbs resulted in greater linear changes in MTsat and $R1$ in the upper arm areas of the cerebellum and the thalamus. For the diffusion tensor imaging results, training of the upper limbs resulted in greater linear changes in fractional anisotropy (FA; red) and axial diffusivity (AD; blue) in the upper limb regions of the corticospinal tracts. Training of the lower limbs resulted in greater linear changes in fractional anisotropy (FA; yellow) and axial diffusivity (AD; magenta), and greater quadratic changes in radial diffusivity (RD; green) in the lower limb regions of the corticospinal tracts.

within ±90 ms of the cue; RT was the mean absolute deviation in ms from the ideal response. Data were analyzed in MATLAB R2016b. Left and right performance was evaluated separately, then pooled if no laterality effects were found. Learning was modeled using an exponential fit, which effectively captures the typical pattern of rapid initial improvement followed by a gradual plateau as performance approaches a maximum. Empirical evidence across thousands of learning series covering a broad range of tasks supports the exponential function as a parsimonious and general law of learning[33], providing a simple framework with interpretable parameters for learning rate and asymptotic performance. The exponential fit yielded parameters describing %CSR or RT improvement ($\delta$), the performance plateau ($\alpha$), and the performance gain or rate of performance improvement ($\gamma$).

**MRI acquisition and processing**

MRI scans were acquired on a 3T Siemens Skyra Fit using two protocols: multiparameter mapping (MPM) for $R1$, MTsat, and $R2^*$, and diffusion MRI (dMRI) for tensor metrics (FA, MD, AD, and RD). The MRI protocol was identical to that described in detail in Azzarito et al.[17], and Emmenegger et al.[16]. MPM enables quantitative assessment of microstructure via the longitudinal relaxation rate ($R1 = 1/T1$), magnetization transfer saturation (MTsat), and effective transverse relaxation rate ($R2^*$). $R1$ and MTsat are commonly used as myelin markers[34–36], whereas $R2^*$ is highly sensitive to local iron content[22,35,37]. Diffusion MRI measures the rate and directionality of water movement[38–40]. Radial diffusivity (RD) is relatively specific to myelin integrity, whereas axial diffusivity (AD) has been associated with the integrity of axonal cytoarchitecture[34,41–49]. FA, which depends on both AD and RD, is sensitive to changes in both axonal and myelin structure[50].

MPM maps were generated using the hMRI toolbox in statistical parametric mapping (SPM12), as described in detail in Azzarito et al.[17] dMRI data were denoised with MRtrix3 and corrected for motion/eddy currents in FSL, as described in Emmenegger et al.[16]. Diffusion tensors were fit using weighted least squares and co-registered to MTsat maps, then spatially normalized to MNI152 space. MPM maps were smoothed with tissue-specific kernels (5 mm full width at half maximum (FWHM) for GM, 3 mm for WM). DTI maps used 5 mm kernels. For voxel-based morphometry, 6 mm (GM) and 4 mm (WM) kernels were applied. Voxels with GM probability <0.2 or WM < 0.6 were excluded. Regions of interest (ROIs) were defined a priori based on prior training studies[17,51] and matched those used previously[17]. They included: sensorimotor cortices, cranial corticospinal tract (CST), thalamus, cerebellum, and the hippocampal formation (hippocampus plus entorhinal cortex, dentate gyrus, cornu ammonis, subiculum). CST and hippocampus ROIs were derived from FSL probabilistic atlases; other regions were defined using the SPM anatomy toolbox. All ROIs were in MNI space and were combined for each hemisphere and tissue type (GM and WM).

## Statistics and reproducibility

As noted above, the total cohort comprised 49 subjects, including healthy males ($n = 32$) and males with chronic SCI ($n = 17$, >6 months post-injury; Table 1). Healthy volunteers and SCI patients were further subdivided into five groups: healthy upper-limb trainees ($n = 9$), healthy lower-limb trainees ($n = 9$), healthy non-trainees ($n = 14$), SCI upper-limb trainees ($n = 9$), and SCI lower-limb trainees ($n = 8$). Grouping was based on motor capacity; patients without residual lower-limb function were excluded from lower-limb training.

Performance, measured as percentage of correct stimulus responses (% CSR) and the response time (RT), and corresponding improvements were fitted using an exponential model ($y = \alpha - \delta e^{(-\gamma t)} + \varepsilon$), where $y$ denotes the training outcome (%CSR or RT), and $t$ represents the training time point. The parameter $\alpha$ corresponds to the asymptotic value, reflecting the performance level reached after extended training (plateau). The parameter $\delta$ quantifies the magnitude of improvement from baseline to the asymptote, while $\gamma$ characterizes the curvature of the learning trajectory, indicating the rate at which performance approaches its asymptotic level. The residual term $\varepsilon$ captures random error and is assumed to follow a normal distribution with a mean of zero. Statistical analyses were conducted in Stata 15.0 to evaluate: (i) improvement over time in SCI trainees, (ii) differences between SCI and healthy trainees, (iii) differences between SCI upper- and lower-limb groups, and (iv) side-to-side differences in SCI trainees. The Shapiro–Wilk test assessed normality. As distributions were non-normal, nonparametric tests were used: a one-sample Wilcoxon test for (i), and Wilcoxon rank-sum tests for (ii–iv). Retention was evaluated by comparing %CSR and RT on days 28 and 84 using Wilcoxon signed-rank tests. SPM12 (v7487) was used to analyze longitudinal structural changes in: (i) SCI trainees vs. healthy non-trainees, (ii) SCI upper- vs. lower-limb trainees, (iii) correlation between brain changes and performance, and (iv) SCI vs. healthy trainees. Models included intercept, linear, and quadratic time terms, plus covariates for age and total intracranial volume (TIV). The linear term was orthogonalized to distinguish sustained from transient changes. Statistical maps were family-wise error (FWE)-corrected at $p < 0.05$, using a cluster-forming threshold of $p < 0.001$ and a minimum cluster size of 20 voxels. Multiple regression in SPM tested associations between behavioral improvements ($\alpha$, $\delta$, and $\gamma$ for %CSR and RT) and voxel-wise structural changes in SCI trainees. Outliers (>3 SD from the mean) were excluded to minimize bias[17].

Differences in training-induced structural changes between upper- and lower-limb SCI trainees were analyzed in SPM by directly comparing linear and quadratic time components. We compared training-induced structural brain changes between SCI and healthy trainees by testing group differences in the linear ($\beta_1$) and quadratic ($\beta_2$) model coefficients within the clusters that exhibited significant responses to training. Both frequentist (RStudio, version 2022.07.1) and Bayesian statistics (JASP, version 0.17.1, with the default prior, Cauchy distribution with a scale parameter of 0.707) were applied. Bayes factors ($BF_{01}$) were used to quantify evidence for the null hypothesis (no group difference) versus the alternative. This approach allowed us to detect significant effects and also assess similarity where appropriate[52].

To evaluate differences in the retention of training-induced microstructural changes between healthy and SCI trainees, the mean values of MRI parameters were first calculated within clusters that exhibited significant responses to training. Frequency and probability testing were then performed to identify differences and assess the likelihood of equivalency within the SCI trainee group between day 28 and day 84. Subsequently, the same statistical approach was applied to compare healthy and SCI trainees on day 84. For this analysis, in addition, paired t-tests were conducted in RStudio, and Bayesian paired and independent t-tests were performed in JASP[52].

## Reporting summary

Further information on research design is available in the Nature Portfolio Reporting Summary linked to this article.

## Data availability

Anonymized data underlying the graphs are provided in the Supplementary Data. Other anonymized data will be made available upon request from qualified investigators.

## Code availability

The analyses were performed primarily using established, GUI-based software packages, including MATLAB R2016b, SPM12 (v7487), the hMRI toolbox, Anatomy Toolbox, FSL, Stata 15.0, JASP (v0.17.1), MRtrix3, and RStudio (v2022.07.1). Support regarding the use of the software packages listed above will be provided upon reasonable request by the corresponding author.

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

## Acknowledgements

We would like to thank all participants for their participation in this study. We thank Eric Reese (https://github.com/kyzentun) for selflessly offering his time and expertise in the writing of the Stepmania scripts. We also thank the staff of the Balgrist Spinal Cord Injury Center for their guidance and support in developing and carrying out this study. This work was supported by Wings for Life, Austria (WFL-CH-007/14). P.F. is funded by a SNF Eccellenza Professorial Fellowship grant (PCEFP3_181362/1). S.M. was funded by the European Union. Views and opinions expressed are those of the author(s) only and do not necessarily reflect those of the European Union or the European Research Council Executive Agency. Neither the European Union nor the granting authority can be held responsible for them. S.M. was supported by an ERC grant (Acronym: MRStain, grant agreement ID: 101089218, DOI: 10.3030/101089218). N.W. has received funding from the European Research Council under the European Union's Seventh Framework programme (FP7/2007-2013)/ERC grant agreement 616905 and from the BMBF (01EW1711A & B) in the framework of ERA-NET NEURON. P.F., N.W. have received funding from the European Union's Horizon 2020 Research and Innovation program (grant agreement no. 681094).

## Author contributions

Tim Max Emmenegger: conceptualization, formal analysis, investigation, methodology, validation, visualization, writing—original draft; Gergely David: conceptualization, formal analysis, methodology, writing—review and editing; Siawoosh Mohammadi: conceptualization, data curation, investigation, methodology, writing—review and editing; Gabriel Ziegler: investigation, methodology, writing—review and editing; Martina Callaghan: conceptualization, investigation, methodology; Alan Thompson: conceptualization, methodology, writing—review and editing; Karl Friston conceptualization, methodology, writing—review and editing); Nikolaus Weiskopf: conceptualization, investigation, methodology, writing—review and editing; Tim Killeen: conceptualization, formal analysis, methodology, project administration, writing—review and editing, and Patrick Freund conceptualization, data curation, funding acquisition, investigation, methodology, project administration, writing—review and editing.

## Competing interests

The authors declare no competing interests.

## Additional information

**Supplementary information** The online version contains Supplementary material available at https://doi.org/10.1038/s42003-026-09793-7.

