## [Transparent Peer Review File · Communications Biology]

Neuronal plasticity during motor rehabilitation training after spinal cord injury

Corresponding Author: Professor Patrick Freund

Version 0:

Reviewer comments:

Reviewer #1

(Remarks to the Author)

In this manuscript, T. M. Emmenegger and collaborators report the effects of a bimanual-bipedal motor training paradigm on brain structural plasticity in cohorts of both uninjured individuals and individuals with SCI. The authors report lasting improvements in performance, reflected in faster response times and higher accuracy across the training period. These behavioral gains were accompanied by structural brain changes, including gray matter/white matter volume alterations and myelin plasticity in key sensorimotor regions. Notably, the study further highlights somatotopic differences in brain reorganization between lower- and upper-limb SCI trainees, which is of particular relevance for the design of personalized therapeutic approaches.

Overall, this work provides valuable longitudinal evidence of motor training-induced brain structural plasticity in SCI, with insights into regional volume and myelination dynamics across different stages of training. The findings are of clear interest to the neuroscience and rehabilitation communities.

Comments and Questions

1. The introduction would benefit from a dedicated section summarizing the current state of knowledge on plasticity following SCI. Specifically, it would be useful to highlight existing evidence on structural plasticity as measured by the MRI metrics employed in this study.
2. Please clarify why only male participants were included. Was this an intentional inclusion criterion? Additionally, were all SCI participants right-handed? Were both limbs trained similarly, or was there a focus on the weaker side? These details are important for understanding the generalizability of the findings and could be added in Table S1.
 - o Some information from Table S1 could be useful to incorporate in the corresponding Results section. The SCI cohort is heterogenous in term of lesion level, severity, and time since injury. Were these factors considered in the analyses performed? Time since injury could influence greatly the difference in performance and the brain changes observed.
 - o To improve clarity, consider representing individual SCI trajectories in figures with distinct shading or lines, ensuring readability is preserved.
3. The manuscript states, "Each session included 15 bouts with 2-minute rests." Please clarify whether the 2-minute rest periods occurred after each bout. If so, the total active training time would amount to 30 minutes per session rather than the reported 60 minutes. Could the authors elaborate on whether this daily training duration can truly be considered intensive?
4. Please provide appropriate references, if any, supporting the use of an exponential fit for modeling learning curves.
5. It appears that no familiarization phase was included prior to training. How do the authors differentiate between improvements due to task learning and those reflecting neurological / functional recovery? Similar to healthy controls, SCI participants improved performance and showed associated brain changes, with performance plateauing during the retention phase. Interestingly, the faster rate of improvement in SCI participants could indicate additional biological mechanisms, which warrant further discussion. Furthermore, differences in brain changes observed across early (0–14 days) versus later (14–28 days) training phases might reflect recovery-related processes beyond motor learning. Drawing direct parallels with prior findings in healthy controls could strengthen the discussion.
6. The section on MRI acquisition would be improved by briefly describing the parameters analyzed (e.g., R1, MT saturation) and their physiological relevance. This would help readers better understand the significance of the reported results.
7. The Methods section refers to "healthy upper-limb trainees" and "healthy lower-limb trainees," but these groups are not mentioned in the Results or Discussion. A comparison between upper- and lower-limb trainees' subgroups in SCI and healthy controls could provide additional insights, particularly regarding the brain regions seen involved and implications for neurological recovery.

8. How do the authors explain the higher baseline performance observed in upper-limb compared to lower-limb SCI trainees? It appears that it was also the case for healthy controls subgroups.

9. The comparison between SCI trainees and healthy non-trainees (Table 1) is difficult to interpret. The reported results may partly reflect baseline group differences at the time of assessment rather than training-specific effects. Please clarify this point. Although challenging to implement, including a SCI non-training control group would have significantly strengthened the study by better isolating training-induced changes. Please discuss as a potential limitation to the study.

10. Please ensure thorough proofreading of the manuscript prior to resubmission, as numerous typographical errors are present throughout. For example:

Line 67: neurologicalconditions

Line 209: thes volumetric changes

Line 216: improvments

Line 245/246: patientns / traineds ...

Reviewer #2

(Remarks to the Author)

Dear Authors, This is a comprehensive well designed, performed, analyzed, and reported study. I think it has exceptional value. The only issue that I would suggest, is a bit of increased clarity relating to similarity or differences in plasticity in the dominant versus non-dominant hand region, line 234. Other than this, excellent work. I think your claims regarding the potential of the motor regions to express significant evidence of neuroplasticity induced by rehabilitative training is an important issue and well supported by your findings. I think these findings will be of great interest to the entire neuroscience community.

Version 1:

Reviewer comments:

Reviewer #1

(Remarks to the Author)

Thank you to the authors for their detailed replies and for providing a revised version of the manuscript. The study addresses an important topic and the results are of clear relevance to the neurorehabilitation field. I have only a few minor suggestions that may be considered at the authors' discretion:

-The specific acquisition parameters for each MRI sequence are not described in the Methods section. Were the parameters identical to those reported in the cited reference? If they differed (even slightly), it would be helpful to specify this.

-The symbols δ , δ and γ should be defined upon first appearance in the Statistical Analysis subsection, rather than only when discussed in the Results.

-The sentence "While this restriction may limit the generalizability of the findings to female populations, the higher prevalence of traumatic spinal cord injury in men²² supports the relevance of the study to the majority of affected patients" could be moved to the Limitations section.

-The authors may wish to discuss potential differences in medication as a factor that could influence outcomes between upper- and lower-limb trainees, and/or include it as a limitation if it was not controlled for.

-Please consider clarifying that the active training task duration was 30 minutes with rests rather than 60 in the discussion.

-In the text, most references appear after punctuation marks. A revision for consistency in reference formatting is recommended.

Table 1: instead of leaving empty cells for healthy controls, the table could present mean \pm SD for age for this group, and "–/NA" for other columns.

Reviewer #1

In this manuscript, T. M. Emmenegger and collaborators report the effects of a bimanual-bipedal motor training paradigm on brain structural plasticity in cohorts of both uninjured individuals and individuals with SCI. The authors report lasting improvements in performance, reflected in faster response times and higher accuracy across the training period. These behavioral gains were accompanied by structural brain changes, including gray matter/white matter volume alterations and myelin plasticity in key sensorimotor regions. Notably, the study further highlights somatotopic differences in brain reorganization between lower- and upper-limb SCI trainees, which is of particular relevance for the design of personalized therapeutic approaches. Overall, this work provides valuable longitudinal evidence of motor training-induced brain structural plasticity in SCI, with insights into regional volume and myelination dynamics across different stages of training. The findings are of clear interest to the neuroscience and rehabilitation communities.

Comments and Questions

R1.1. The introduction would benefit from a dedicated section summarizing the current state of knowledge on plasticity following SCI. Specifically, it would be useful to highlight existing evidence on structural plasticity as measured by the MRI metrics employed in this study.

We have now expanded the Introduction to offer a more detailed description of neuroplasticity responses after SCI. The corresponding passage now reads as follows:

“More specifically, evidence indicates that SCI triggers significant neuroplasticity across multiple levels of the nervous system, offering potential for functional recovery. In particular, studies have demonstrated that spinal neural circuitry exhibits remarkable automaticity and plasticity following injury, with the capacity to perform complex motor tasks despite disrupted supraspinal input.⁸ Plasticity occurs at cerebral cortical, brainstem, spinal cord, and peripheral levels, with mechanisms varying by timeframe: acute changes involve unmasking latent synapses through neurotransmitter modulation, while chronic adaptations include synaptic efficacy changes and axonal sprouting.⁹ Recent systematic review evidence confirms that SCI patients experience plastic changes both spontaneously and through specific neurorehabilitation training, with exercise rehabilitation promoting WM plasticity and increased myelin content.¹⁰ These plasticity changes have been observed in motor and non-motor learning brain regions^{7,11,12} and were quantified using metrics such as myelin water fraction,¹¹ fractional anisotropy,¹² and volumetric analyses.⁷ However, although training-induced volumetric neuroplastic changes have been reported,^{6,7} microstructural changes in myelin or axonal integrity, as quantified by advanced quantitative MRI (qMRI) techniques, remain rarely explored.”

6. Burciu RG, Fritsche N, Granert O, et al. Brain Changes Associated with Postural Training in Patients with Cerebellar Degeneration: A Voxel-Based Morphometry Study. *Journal of*

7. Villiger M, Grabher P, Hepp-Reymond MC, et al. Relationship between structural brainstem and brain plasticity and lower-limb training in spinal cord injury: a longitudinal pilot study. *Front Hum Neurosci.* 2015;9:1-10. doi:10.3389/fnhum.2015.00254

8. Edgerton VR, Tillakaratne NJK, Bigbee AJ, de Leon RD, Roy RR. PLASTICITY OF THE SPINAL NEURAL CIRCUITRY AFTER INJURY. *Annu Rev Neurosci.* 2004;27(1):145-167. doi:10.1146/annurev.neuro.27.070203.144308
9. Ding Y, Kastin A, Pan W. Neural Plasticity After Spinal Cord Injury. *Curr Pharm Des.* 2005;11(11):1441-1450. doi:10.2174/1381612053507855
10. Calderone A, Cardile D, De Luca R, Quartarone A, Corallo F, Calabrò RS. Brain Plasticity in Patients with Spinal Cord Injuries: A Systematic Review. *Int J Mol Sci.* 2024;25(4):2224. doi:10.3390/ijms25042224
11. Faw TD, Lakhani B, Schmalbrock P, et al. Eccentric rehabilitation induces white matter plasticity and sensorimotor recovery in chronic spinal cord injury. *Exp Neurol.* 2021;346:113853. doi:10.1016/j.expneurol.2021.113853
12. Seáñez-González I, Pierella C, Farshchiansadegh A, et al. Body-Machine Interfaces after Spinal Cord Injury: Rehabilitation and Brain Plasticity. *Brain Sci.* 2016;6(4):61. doi:10.3390/brainsci6040061

R1.2. Please clarify why only male participants were included. Was this an intentional inclusion criterion?

It was an intentional inclusion criterion, and we have now clarified this point in the "Participants" section as the follows:

"Recruitment was limited to males to avoid potential sex-related confounding effects on neuroplastic processes, which may differ in temporal dynamics, spatial distribution, or magnitude. While this restriction may limit the generalizability of the findings to female populations, the higher prevalence of traumatic spinal cord injury in men²² supports the relevance of the study to the majority of affected patients."

22. Jackson AB, Dijkers M, DeVivo MJ, Poczatek RB. A demographic profile of new traumatic spinal cord injuries: Change and stability over 30 years. *Arch Phys Med Rehabil.* 2004;85(11):1740-1748. doi:10.1016/j.apmr.2004.04.035

R1.3. Additionally, were all SCI participants right-handed?

Some SCI participants showed a preference for the left hand, which is now stated in the *Participants* section:

"This study included 32 healthy males (23–62 years) and 17 males with chronic SCI (23–70 years, >6 months post-injury; Table 1). All healthy participants and most patients were right handed, while four SCI participants showed a preference for the left hand, as assessed with the Edinburgh Handedness Inventory,²¹ but this was likely affected by post-injury preferences on account of asymmetrical limb function."

21. Oldfield RC. The assessment and analysis of handedness: The Edinburgh inventory. *Neuropsychologia.* 1971;9(1):97-113. doi:10.1016/0028-3932(71)90067-4

R.1.4 Were both limbs trained similarly, or was there a focus on the weaker side? These details are important for understanding the generalizability of the findings and could be added in Table S1.

We have clarified this point in the "Motor skill learning" section as the follows:

"The total training time was designed to be equal for the left and right sides; however, participants were free to compensate by using their stronger side for most inputs. Although this was not explicitly measured, it was likely to be the case in some instances of highly unilateral injuries."

R.1.5. Some information from Table S1 could be useful to incorporate in the corresponding Results section. The SCI cohort is heterogenous in term of lesion level, severity, and time since injury. Were these factors considered in the analyses performed?

We have now moved Table S1 into the main manuscript (new Table 1) to provide detailed cohort information to the reader without the need to access the supplementary data. Furthermore, we have included additional information, such as injury severity, injury completeness and time since injury, to the Demographics section as follows:

"Among SCI trainees, AIS grades (A–D) were similarly distributed across upper and lower limb subgroups (Fisher's exact test: $p=0.390$), as were injury completeness (Fisher's exact test: $p = 0.294$) and time since injury (Welch's t-test: $p = 0.774$, Table 1). However, the SCI trainees differed significantly regarding lesion level (Welch's t-test: $p = 0.008$), with more tetraplegic participants in the upper limb group compared to the lower limb subgroup (Table 1)."

Regarding the inclusion of such factors in the statistical analyses: given our relatively small sample size (as stated in the Limitations section) and the fact that all patients were in the chronic stage, we chose not to include these factors as covariates to avoid potential overfitting.

Furthermore, no significant differences were found in severity (as measured by AIS grades or completeness), time since injury, or age. As group allocation was based on motor capacity as mentioned in the "Participants section", lesion level differed significantly between the two groups, with higher lesions observed in the upper-limb trainees. Nevertheless, we agree that these factors could have potentially influenced baseline differences; however, at baseline we only found that upper-limb trainees had higher %CSR than lower-limb trainees, which is in line with the behavioral baseline differences observed in healthy volunteers.¹⁷ In terms of improvement (δ) or rate of change (γ), we did not detect any significant differences between the SCI subgroups. Furthermore, to minimize baseline influences, all other investigations, such as the assessment of somatotopic differences, focused only on linear or quadratic changes during the relatively short time frame of 28 days, in relation to the mean time since injury (95 ± 85 months).

To address this limitation, we have expanded the Limitations section as follows:

"A further issue to consider is that the SCI upper- and lower-limb trainees showed significant differences in lesion level, with upper-limb trainees having higher lesion levels compared to lower-

limb trainees. This group allocation was intentional to achieve comparable motor capacity in both groups for their respective training tasks. This baseline difference could nevertheless have influenced behavioral performance, as a higher %CSR was found in SCI upper-limb trainees compared to SCI lower-limb trainees at baseline, without a significant difference in RT. We note, however, that this baseline difference was also observed in healthy individuals.²³ Furthermore, it cannot be excluded that differences in lesion level may have concealed some of the somatotopic neuroplasticity differences, as higher lesion levels have been shown to lead to greater neurodegeneration compared to lower lesion levels,^{1,60,61} potentially counteracting neuroplastic changes to varying degrees.”

1 Emmenegger TM, Pfyffer D, Curt A, et al. Longitudinal motor system changes from acute to chronic spinal cord injury. *Eur J Neurol.* 2024;(October 2023):1-12. doi:10.1111/ene.16196

23 Azzarito M, Emmenegger TM, Ziegler G, et al. Coherent, time-shifted patterns of microstructural plasticity during motor-skill learning. *Neuroimage.* 2023;274(April):120128. doi:10.1016/j.neuroimage.2023.120128

60. Schading S, David G, Emmenegger TM, et al. Dynamics of Progressive Degeneration of Major Spinal Pathway Following Spinal Cord Injury: A Longitudinal Study. *SSRN Electronic Journal.* 2022;14(2):109-117. doi:10.2139/ssrn.4248699

61. Azzarito M, Seif M, Kyathanahally S, Curt A, Freund P. Tracking the neurodegenerative gradient after spinal cord injury. *Neuroimage Clin.* 2020;26(October 2019):102221. doi:10.1016/j.nicl.2020.102221

R1.6. Time since injury could influence greatly the difference in performance and the brain changes observed.

We agree that time since injury could potentially influence both performance and brain changes. However, in this chronic cohort, we did not observe any significant differences between the SCI subgroups in terms of time since injury ($p = 0.774$). Furthermore, a post hoc analysis did not find any significant relationships between time since injury, performance parameters, and brain changes ($p > 0.25$, even when including the outliers; see plots below for behavioral parameters and time since injury).

The figure above shows the association between time since injury and the performance parameters (α = plateau, δ = improvement, and γ = improvement speed) of percent correct stimulus responses (%CSR) and response time (RT), including subjects exceeding 3 standard deviations from the mean.

To address this limitation, we have made the following statement to the Limitations section:

“Our study has limitations. The cohort was small (17 male patients) and limited to patients in the chronic phase, reducing generalizability to females, acute injuries, or other severities.”

R1.7. To improve clarity, consider representing individual SCI trajectories in figures with distinct shading or lines, ensuring readability is preserved.

We have added distinct shading in Figures 1, 2, and 3, and included the healthy control data as a reference for Figure 1 and Figure 2. Please see the updated figures below.

The figure caption for Figure 1 has been updated as follows:

“Experimental design, training task and performance improvements data. The experimental design (A) included MRI acquisition and training assessments at baseline (day 0) during the training period (days 7, 14 and 28) and at final retention assessment (day 84). Sixty minutes of supervised training in a motor-skill task was undertaken four times per week for 4 consecutive weeks (B), whereby participants were required to activate inputs with their hands or feet (depending on whether they were allocated to the upper (C) or lower limb (D) training groups) in response to rhythmic aural and visual stimuli in the dance game Stepmania. The participant was tasked with selecting and activating the correct symbol at the precise moment the scrolling arrow overlapped with a set of static arrows at the top of the screen. Behavioural improvement, defined as the percentage of correct stimulus responses (%CSR), and response time (RT; the deviation in ms from the ideal response) were measured during a formal, standardised training assessment at weekly intervals (see methods). Spinal cord injured and healthy trainees values for these metrics are plotted as dots (blue: healthy trainees; red: spinal cord injured trainees; E: n=35 and F: upper limb n=18, lower limb n=17), while participant-specific behavioural curves were computed (thin lines) along with the group median of the plotted individuals (red line). Note that the results from the healthy controls trainees (blue lines) have been reported previously (Azzarito et al., 2023) and are shown for illustrative purposes.”

For Figure 2, the color shading has been adjusted to align with the color scheme used in Figure 1. The figure caption has been updated accordingly and now reads as follows:

“Figure 2: Training-induced changes during the learning of the motor skill task, combining all spinal cord injured (SCI) upper and lower limb trainees (in red, n=17), compared to healthy controls trainees (in blue, n=18) and healthy non-trainees (in green, n=14). The black line indicates the differences between SCI trainees and non-trainees (SCI trainees – Non-trainees). Dashed lines represent periods without training. On the left column, micro-macrostructural changes are depicted with myelin-sensitive magnetization transfer (MTsat, in red), longitudinal relaxation rate (R1, in yellow), and volume metric changes (in green). On the right column, diffusional changes are illustrated with fractional anisotropy (FA, in yellow), mean diffusivity (MD, in blue), axial diffusivity (AD, in red), and radial diffusivity (RD, in green). The bold blue and green lines represent results from

healthy trainees, reported previously (Azzarito et al., 2023; Emmenegger et al., 2024) and are shown for illustrative purposes.”

The figure caption for Figure 3 has been updated accordingly and now reads as follows:

“Associations between linear (β_1) or quadratic (β_2) micro- and macrostructural changes (left column: MT_{sat} = magnetization transfer saturation and volume) or diffusion parameters (right column: FA = fractional anisotropy, and RD = radial diffusivity) and performance improvements parameters (response time = RT , and percentage of correct stimulus responses = $\%CSR$, α = plateau, δ = improvement, and γ = improvement speed) of spinal cord injury trainees. $n=16$ ”

R1.8. The manuscript states, "Each session included 15 bouts with 2-minute rests." Please clarify whether the 2-minute rest periods occurred after each bout. If so, the total active training time would amount to 30 minutes per session rather than the reported 60 minutes. Could the authors elaborate on whether this daily training duration can truly be considered intensive?

Thank you very much for pointing this out. To eliminate potential misunderstandings, we have now clarified this in the “Motor Skill Learning” section:

“Training lasted four weeks, with 60-minute sessions (including breaks) four times per week (Figure 1A). No practice occurred between day 28 and the follow-up on day 84. StepMania 5 (Beta 3) was used for training, with custom input devices based on the trained limb. Lower-limb training used a dance mat (Impact Dance Platform) connected to a monitor; SCI patients unable to stand wore a ceiling-mounted harness. Upper-limb training used a tabletop interface; participants responded with left and right hands to directional cues on a laptop. Each session consisted of 15 bouts, each lasting 2 minutes, totaling 30 minutes of training, with roughly 2-minute breaks between bouts. During each bout, arrows scrolled to music, and participants responded by triggering inputs when the arrows reached a fixed target. ”

R1.9. Please provide appropriate references, if any, supporting the use of an exponential fit for modeling learning curves.

We have now provided justification for using an exponential fit to model learning curves (“Motor skill learning” section, lines):

“Learning was modeled using an exponential fit, which effectively captures the typical pattern of rapid initial improvement followed by a gradual plateau as performance approaches a maximum. Empirical evidence across thousands of learning series covering a broad range of tasks supports the exponential function as a parsimonious and general law of learning,²⁵ providing a simple framework with interpretable parameters for learning rate and asymptotic performance.”

We also added the following section to the Limitations:

“Furthermore, learning was modeled using an exponential fit, which may not always be the most appropriate model for every individual (Fig. 1E,F). Therefore, to avoid bias from such cases, behavioral outliers were excluded from the statistical analyses. This applied, for example, to a patient in the lower-limb training group for whom an exponential fit produced unphysical values for RT (see Fig. 1E,F).”

25. Heathcote A, Brown S, Mewhort DJK. The power law repealed: The case for an exponential law of practice. *Psychon Bull Rev.* 2000;7(2):185-207. doi:10.3758/BF03212979

R1.10. It appears that no familiarization phase was included prior to training. How do the authors differentiate between improvements due to task learning and those reflecting neurological / functional recovery? Similar to healthy controls, SCI participants improved performance and showed associated brain changes, with performance plateauing during the retention phase. Interestingly, the faster rate of improvement in SCI participants could indicate additional biological mechanisms, which warrant further discussion. Furthermore, differences in brain changes observed across early (0–14 days) versus later (14–28 days) training phases might reflect recovery-related processes beyond motor learning. Drawing direct parallels with prior findings in healthy controls could strengthen the discussion.

We are grateful for these insightful comments regarding the potential effects of familiarization, additional biological mechanisms, and brain changes observed across early (0–14 days) versus later (14–28 days). These questions cannot be fully addressed with the current study design, and further investigations will be necessary. We now discuss these points in the manuscript as follows:

“To some degree, particularly at the beginning of the training period, the performance gains could also be attributed to familiarization processes.”

“Nevertheless, further investigations are needed to clarify whether the underlying biological processes causing the observed alterations in R1, MTsat, R2, FA, MD, AD, or RD are identical to those in healthy controls, or whether additional or more pronounced biological processes are responsible for these findings. It should be noted that the participating SCI patients had been living with their injury for an average of 95 months, and the observed neuroplastic changes followed a similar pattern to those reported in healthy controls. Specifically, an initial decrease (~0–14 days) followed by an increase (~14–28 days) in MT, R1, and R2* has been described in healthy individuals,¹⁸ along with linear increases in FA and AD and decreases in RD and MD.¹⁷ These patterns were also observed in the SCI patients, suggesting that pathological processes, typically described as mainly linear and occurring in the opposite direction,^{1,56} may not have played a substantial role in the present findings. Whether the initial and subsequent phases should be viewed as one continuous process or as two distinct and independent processes requires further investigation.”*

1. Emmenegger TM, Pfyffer D, Curt A, et al. Longitudinal motor system changes from acute to chronic spinal cord injury. *Eur J Neurol.* 2024;(October 2023):1-12. doi:10.1111/ene.16196

17. Emmenegger T, David G, Mohammadi S, et al. Temporal dynamics of white and gray matter plasticity during motor skill acquisition: a comparative diffusion tensor imaging and multiparametric mapping analysis. *Cerebral Cortex.* 2024;34(8). doi:10.1093/cercor/bhae344

18. Azzarito M, Emmenegger TM, Ziegler G, et al. Coherent, time-shifted patterns of microstructural plasticity during motor-skill learning. *Neuroimage*. 2023;274(April):120128. doi:10.1016/j.neuroimage.2023.120128

56. Ilvesmäki T, Koskinen E, Brander A, Luoto T, Öhman J, Eskola H. Spinal cord injury induces widespread chronic changes in cerebral white matter. *Hum Brain Mapp*. 2017;38(7):3637-3647. doi:10.1002/hbm.23619

R1.11. The section on MRI acquisition would be improved by briefly describing the parameters analyzed (e.g., R1, MT saturation) and their physiological relevance. This would help readers better understand the significance of the reported results.

We have added a few sentences describing the physiological relevance of the MRI parameters ("MRI acquisition and processing" section).

"MPM enables quantitative assessment of microstructure via the longitudinal relaxation rate ($R1 = 1/T1$), magnetization transfer saturation (MTsat), and effective transverse relaxation rate ($R2^$). $R1$ and MTsat are commonly used as myelin markers,²⁶⁻²⁸ whereas $R2^*$ is highly sensitive to local iron content.^{27,29,30} Diffusion MRI measures the rate and directionality of water movement.³¹⁻³³ Radial diffusivity (RD) is relatively specific to myelin integrity, whereas axial diffusivity (AD) has been associated with the integrity of axonal cytoarchitecture.^{26,34-42} Fractional anisotropy (FA), which depends on both AD and RD, is sensitive to changes in both axonal and myelin structure.⁴³"*

26. Georgiadis M, Schroeter A, Gao Z, et al. Nanostructure-specific X-ray tomography reveals myelin levels, integrity and axon orientations in mouse and human nervous tissue. *Nat Commun*. 2021;12(1):2941. doi:10.1038/s41467-021-22719-7

27. Natu VS, Gomez J, Barnett M, et al. Apparent thinning of human visual cortex during childhood is associated with myelination. *Proc Natl Acad Sci U S A*. 2019;116(41):20750-20759. doi:10.1073/pnas.1904931116

28. Schmierer K, Scaravilli F, Altmann DR, Barker GJ, Miller DH. Magnetization transfer ratio and myelin in postmortem multiple sclerosis brain. *Ann Neurol*. 2004;56(3):407-415. doi:10.1002/ana.20202

29. Ghadery C, Pirpamer L, Hofer E, et al. $R2^*$ mapping for brain iron: associations with cognition in normal aging. *Neurobiol Aging*. 2015;36(2):925-932. doi:10.1016/j.neurobiolaging.2014.09.013

30. Langkammer C, Krebs N, Goessler W, et al. Quantitative MR Imaging of Brain Iron: A Postmortem Validation Study. *Radiology*. 2010;257(2):455-462. doi:10.1148/radiol.10100495

31. Basser PJ, Mattiello J, LeBihan D. MR diffusion tensor spectroscopy and imaging. *Biophys J*. 1994;66(1):259-267. doi:10.1016/S0006-3495(94)80775-1

32. Seiler A, Nöth U, Hok P, et al. Multiparametric Quantitative MRI in Neurological Diseases. *Front Neurol.* 2021;12(March). doi:10.3389/fneur.2021.640239
33. Chen Y, Baraz J, Xuan SY, et al. Multiparametric Quantitative MRI of Peripheral Nerves in the Leg: A Reliability Study. *Journal of Magnetic Resonance Imaging.* 2024;59(2):563-574. doi:10.1002/jmri.28778
34. Song SK, Sun SW, Ramsbottom MJ, Chang C, Russell J, Cross AH. Dysmyelination revealed through MRI as increased radial (but unchanged axial) diffusion of water. *Neuroimage.* 2002;17(3):1429-1436. doi:10.1006/nimg.2002.1267
35. Song SK, Sun SW, Ju WK, Lin SJ, Cross AH, Neufeld AH. Diffusion tensor imaging detects and differentiates axon and myelin degeneration in mouse optic nerve after retinal ischemia. *Neuroimage.* 2003;20(3):1714-1722. doi:10.1016/j.neuroimage.2003.07.005
36. Sun SW, Liang HF, Trinkaus K, Cross AH, Armstrong RC, Song SK. Noninvasive detection of cuprizone induced axonal damage and demyelination in the mouse corpus callosum. *Magn Reson Med.* 2006;55(2):302-308. doi:10.1002/mrm.20774
37. Sun SW, Liang HF, Cross AH, Song SK. Evolving Wallerian degeneration after transient retinal ischemia in mice characterized by diffusion tensor imaging. *Neuroimage.* 2008;40(1):1-10. doi:10.1016/j.neuroimage.2007.11.049
38. Budde MD, Kim JH, Liang HF, et al. Toward accurate diagnosis of white matter pathology using diffusion tensor imaging. *Magn Reson Med.* 2007;57(4):688-695. doi:10.1002/mrm.21200
39. Kim JH, Loy DN, Liang HF, Trinkaus K, Schmidt RE, Song SK. Noninvasive diffusion tensor imaging of evolving white matter pathology in a mouse model of acute spinal cord injury. *Magn Reson Med.* 2007;58(2):253-260. doi:10.1002/mrm.21316
40. Zhang J, Jones M, DeBoy CA, et al. Diffusion tensor magnetic resonance imaging of Wallerian degeneration in rat spinal cord after dorsal root axotomy. *J Neurosci.* 2009;29(10):3160-3171. doi:10.1523/JNEUROSCI.3941-08.2009
41. Xie M, Wang Q, Wu TH, Song SK, Sun SW. Delayed axonal degeneration in slow Wallerian degeneration mutant mice detected using diffusion tensor imaging. *Neuroscience.* 2011;197:339-347. doi:10.1016/j.neuroscience.2011.09.042
42. Brennan FH, Cowin GJ, Kurniawan ND, Ruitenberg MJ. Longitudinal assessment of white matter pathology in the injured mouse spinal cord through ultra-high field (16.4T) in vivo diffusion tensor imaging. *Neuroimage.* 2013;82:574-585. doi:10.1016/j.neuroimage.2013.06.019
43. Beaulieu C. What Makes Diffusion Anisotropic in the Nervous System? In: Jones DK, ed. *Diffusion MRI Theory, Methods, and Applications.* Oxford University Press; 2011:92-109.

R1.12. The Methods section refers to "healthy upper-limb trainees" and "healthy lower-limb trainees," but these groups are not mentioned in the Results or Discussion. A comparison between upper- and lower-limb trainees' subgroups in SCI and healthy controls could provide

additional insights, particularly regarding the brain regions seen involved and implications for neurological recovery.

The analysis comparing healthy upper-limb trainees with healthy lower-limb trainees has already been conducted in the past using the MPM¹⁷ and DTI¹⁸ protocols, showing comparable patterns to those observed in the SCI upper- vs lower-limb analysis. Nevertheless, we have now performed additional analyses to assess potential differences between the healthy and SCI upper- and lower-limb trainee groups. These findings have now been added to Supplementary Table 6, and the discussion section has been expanded accordingly as follows:

“We further explored whether there are differences between the upper-limb SCI trainees and healthy upper-limb trainees, as well as between the lower-limb SCI trainees and healthy lower-limb trainees. We found evidence for greater responses in the SCI patients (see Supplementary Table 6); however, changes in the opposite direction were also observed, potentially reflecting different adaptations in mastering the training task between SCI patients and healthy trainees.”

“We found evidence of somatotopic differentiation in arm-associated regions of the CST and sensorimotor cortex compared to lower-limb training,⁵⁷ similar to what has been reported in healthy controls when upper- and lower-limb training was compared.^{17,18”}

18. Azzarito M, Emmenegger TM, Ziegler G, et al. Coherent, time-shifted patterns of microstructural plasticity during motor-skill learning. *Neuroimage*. 2023;274(April):120128. doi:10.1016/j.neuroimage.2023.120128

17. Emmenegger T, David G, Mohammadi S, et al. Temporal dynamics of white and gray matter plasticity during motor skill acquisition: a comparative diffusion tensor imaging and multiparametric mapping analysis. *Cerebral Cortex*. 2024;34(8). doi:10.1093/cercor/bhae344

57. Lemon RN, Morecraft RJ. The evidence against somatotopic organization of function in the primate corticospinal tract. *Brain*. 2023;146(5):1791-1803. doi:10.1093/brain/awac496

R1.13. How do the authors explain the higher baseline performance observed in upper-limb compared to lower-limb SCI trainees? It appears that it was also the case for healthy controls subgroups.

We have now added a possible explanation to the manuscript:

“At baseline, healthy individuals¹⁸ as well as SCI upper-limb trainees showed a greater %CSR compared to their lower-limb counterparts. We argue that because upper limbs generally have higher dexterity than lower limbs, participants using their upper limbs achieved better performance on the same baseline tasks.”

R1.14. The comparison between SCI trainees and healthy non-trainees (Table 1) is difficult to interpret. The reported results may partly reflect baseline group differences at the time of assessment rather than training-specific effects. Please clarify this point. Although challenging to implement, including a SCI non-training control group would have significantly strengthened the study by better isolating training-induced changes. Please discuss as a potential limitation to the study.

We acknowledge that baseline differences between healthy controls and SCI patients exist and have reported them in Supplementary Table 1 and Supplementary Figure 1. To minimize potential influences of such baseline differences, we analyzed the linear and quadratic components while including the intercept in the model. Furthermore, only chronic SCI patients were included, and the study focused exclusively on short-term changes over a 28-day training period. Known pathological processes in SCI (e.g., axonal degeneration and demyelination) would be expected to lead to changes in the opposite direction of those described in the present study. Therefore, we would anticipate that including an SCI non-training control group would have resulted in equal or even greater between-group differences. To illustrate potential influences of baseline differences, we have now added an additional supplementary figure (Supplementary Figure 2) showing the overlapping and adjacent clusters between significant linear and quadratic changes and regions showing significant intercept differences between patients and healthy controls.

Accordingly, we have expanded the Limitations section as follows:

“Nevertheless, we also detected plastic changes, albeit smaller, in some regions that had shown atrophy or demyelination due to the injury (Supplementary Figure 2), a phenomenon that has also been hinted at in stroke recovery and other neurological conditions.^{6,58,59} The observed structural remodeling within chronically affected circuits suggests that rehabilitative training can engage adaptive mechanisms beyond fully spared pathways. Nevertheless, it should be considered that further investigations focusing on these cases are needed to determine whether different biological processes occur in these brain regions compared to others. This is particularly important since the SCI trainees were compared to healthy controls rather than to a non-training chronic SCI group, making it difficult to draw definitive conclusions about the specific biological processes within these regions. However, it could be assumed that equal or even greater differences in qMRI metrics might be observed, as the known pathological processes in SCI (e.g., axonal degeneration and demyelination)^{1,56} would counteract the suggested biological processes such as axonal sprouting or increased myelination.^{5”}

1. Emmenegger TM, Pfyffer D, Curt A, et al. Longitudinal motor system changes from acute to chronic spinal cord injury. *Eur J Neurol*. 2024;(October 2023):1-12. doi:10.1111/ene.16196

5. Zatorre RJ, Fields RD, Johansen-Berg H. Plasticity in gray and white: neuroimaging changes in brain structure during learning. *Nat Neurosci*. 2012;15(4):528-536. doi:10.1038/nn.3045

6. Burciu RG, Fritsche N, Granert O, et al. Brain Changes Associated with Postural Training in Patients with Cerebellar Degeneration: A Voxel-Based Morphometry Study. *Journal of Neuroscience*. 2013;33(10):4594-4604. doi:10.1523/JNEUROSCI.3381-12.2013
56. Ilvesmäki T, Koskinen E, Brander A, Luoto T, Öhman J, Eskola H. Spinal cord injury induces widespread chronic changes in cerebral white matter. *Hum Brain Mapp*. 2017;38(7):3637-3647. doi:10.1002/hbm.23619
58. Gauthier L V., Taub E, Perkins C, Ortmann M, Mark VW, Uswatte G. Remodeling the Brain. *Stroke*. 2008;39(5):1520-1525. doi:10.1161/STROKEAHA.107.502229
59. Sehm B, Taubert M, Conde V, et al. Structural brain plasticity in Parkinson's disease induced by balance training. *Neurobiol Aging*. 2014;35(1):232-239. doi:10.1016/j.neurobiolaging.2013.06.021

R1.15. Please ensure thorough proofreading of the manuscript prior to resubmission, as numerous typographical errors are present throughout. For example:

Line 67: neurologicalconditions

Line 209: thes volumetric changes

Line 216: improvments

Line 245/246: patientns / trainneds ...

We thank the reviewer for pointing out these errors, which we have corrected now.

Reviewer #2 (Remarks to the Author):

R2.1. Dear Authors, This is a comprehensive well designed, performed, analyzed, and reported study. I think it has exceptional value. The only issue that I would suggest, is a bit of increased clarity relating to similarity or differences in plasticity in the dominant versus non-dominant hand region, line 234. Other than this, excellent work. I think your claims regarding the potential of the motor regions to express significant evidence of neuroplasticity induced by rehabilitative training is an important issue and well supported by your findings. I think these findings will be of great interest to the entire neuroscience community.

Thank you very much for appreciating the our work.

Regarding the comment on "differences in plasticity between the dominant and non-dominant hand regions," we acknowledge that this aspect was not explicitly addressed in the current study. Interestingly, more clusters of linear/quadratic volumetric or microstructural changes were observed in the left compared to the right hemisphere, which may be related to the lateralization of handedness, as most participants were right-handed (as mentioned in R1.3). Nevertheless, further investigations focusing on the effects of handedness would be valuable to determine whether, in a population consisting mainly or exclusively of left-handed individuals, a corresponding tendency toward higher number or greater extent of linear/quadratic volumetric or microstructural changes in the right hemisphere would be observed.

Accordingly, we have now modified the discussion as follows:

“Furthermore, there are some indications of a higher prevalence of significant clusters of linear or quadratic volumetric or microstructural changes in the left hemisphere, which could be related to the higher number of right-handed participants. However, whether this asymmetry is exclusively due to handedness, preferential use of the dominant hand, or other factors still needs to be determined.”

Dear Dr. Bessieres and Reviewers,

We appreciate your review of our manuscript and the useful comments provided. All points have been addressed, and the manuscript has been revised accordingly to improve clarity where needed.

Below, we provide our point-by-point responses to the reviewers' comments.

Reviewer #1

Thank you to the authors for their detailed replies and for providing a revised version of the manuscript. The study addresses an important topic and the results are of clear relevance to the neurorehabilitation field. I have only a few minor suggestions that may be considered at the authors' discretion:

1.1-The specific acquisition parameters for each MRI sequence are not described in the Methods section. Were the parameters identical to those reported in the cited reference? If they differed (even slightly), it would be helpful to specify this.

The acquisition parameters were identical to those reported in the cited publications, with no modifications. To clarify this point, we have revised the Methods section to explicitly state:

"The MRI protocol was identical to that described in detail in Azzarito et al.¹⁷ and Emmenegger et al.¹⁶."

1.2-The symbols α , δ and γ should be defined upon first appearance in the Statistical Analysis subsection, rather than only when discussed in the Results.

We appreciate this helpful suggestion. The symbols α , δ , and γ are now defined at their first occurrence in the Motor skill learning subsection of the Methods. Specifically, we introduced the following sentence:

"The exponential fit yielded parameters describing %CSR or RT improvement (δ), the performance plateau (α), and the performance gain or rate of performance improvement (γ)."

1.3-The sentence "While this restriction may limit the generalizability of the findings to female populations, the higher prevalence of traumatic spinal cord injury in men²² supports the relevance of the study to the majority of affected patients" could be moved to the Limitations section.

We have moved this sentence to the Limitations section.

1.4-The authors may wish to discuss potential differences in medication as a factor that could influence outcomes between upper- and lower-limb trainees, and/or include it as a limitation if it was not controlled for.

As medication use was neither controlled for nor systematically assessed in this study, we cannot make specific statements regarding its potential influence on outcomes in upper- versus lower-limb trainees. We have therefore added this aspect as a limitation in the manuscript.

“Furthermore, medication use was not assessed or controlled for and may have influenced the observed differences between patient subgroups as well as comparisons with healthy controls.”

1.5-Please consider clarifying that the active training task duration was 30 minutes with rests rather than 60 in the discussion.

We have clarified in the Discussion that each session comprised approximately 30 minutes of active task performance within a 60-minute training session:

“The persistence of structural changes at the 2-month follow-up (in the absence of further training) implies that relatively short, intensive rehabilitation —consisting of approximately 30 minutes of active task performance within a 60-minute training session— can induce lasting neural reorganization.”

1.6-In the text, most references appear after punctuation marks. A revision for consistency in reference formatting is recommended.

This has been checked and revised to comply with the journal’s reference formatting guidelines.

1.7-Table 1: instead of leaving empty cells for healthy controls, the table could present mean ± SD for age for this group, and “-/NA” for other columns.

Table 1 has been revised to include mean ± SD values where available and “-/NA” where values were not applicable or not available.